

**Quasi-10-day wave activity in the southern high-latitude MLT**
**region and its relation to the large-scale instability and**
**gravity wave drag**
Wonseok Lee[1], In-Sun Song[1], Byeong-Gwon Song[1], Yong Ha Kim[2]
[1]Department of Atmospheric Sciences, Yonsei University, Seoul 03722, South Korea
[2]Department of Astronomy and Space Science, Chungnam National University,
Daejeon 34134, South Korea
*Correspondence to*: In-Sun Song (songi@yonsei.ac.kr)





**Abstract.** Seasonal variation of westward-propagating quasi-10-day wave (Q10DW) in
the mesosphere and lower thermosphere of the Southern Hemisphere (SH) high-latitude
regions is investigated using meteor radar (MR) observations for the period of 2012–
2016 and Specified Dynamics (SD) version of the Whole Atmosphere Community
Climate Model (WACCM). The phase difference of meridional winds measured by two
MRs located in Antarctica gives observational estimates of the amplitude and phase of
Q10DW with zonal wavenumber 1 (W1). The amplitude of the observed Q10DW-W1 is
large around equinoxes. In order to elucidate the variations of the observed Q10DW-W1
and its possible amplification mechanism, we carry out two SD-WACCM experiments
nudged towards the MERRA-2 reanalysis from the surface up to ~60 km (EXP60) and
~75 km (EXP75). Results of the EXP75 indicate that the observed Q10DW-W1 can be
amplified around the barotropic/baroclinic instability regions in the middle mesosphere
around 60°S–70°S. In the EXP60, it is also found that Q10DW-W1 is amplified around
the instability regions, but the amplitude is too large compared with MR observations.
The large-scale instability in the EXP60 in the SH summer mesosphere is stronger than
that in the EXP75 and Microwave Limb Sounder observation. The larger instability in
the EXP60 is related to the large meridional and vertical variations of polar mesospheric
zonal winds in associated with gravity wave parameterization (GWP). Given
uncertainties inherent in GWP, these results can suggest that it is possible for models to
spuriously generate traveling planetary waves such as Q10DW, especially in summer,
due to the excessively strong large-scale instability in the SH high-latitude mesosphere.



## 1 Introduction

A series of Rossby normal modes (free oscillations) is the homogeneous solution
of the governing equations on a sphere linearized with respect to the isothermal and
quiescent reference atmosphere (e.g., Andrews et al., 1987; Forbes et al., 1995; Salby,
1984). Traveling normal modes exhibit clear planetary-scale spatiotemporal oscillations
throughout the whole atmosphere, and for sufficiently large amplitudes, these traveling
planetary waves (PWs) can play an important role in the momentum and energy transfer
to the mean flow (Salby, 1984). Three gravest traveling normal modes have been
observed: Westward-propagating zonal-wavenumber-1 PWs with periods of
approximately 5, 10, and 16 days. The classical wave theory based on the isothermal
and quiescent atmosphere gives the theoretical periods of 5, 8.3, and 12.5 day, but the
periods in the real atmosphere can be shifted to values close to 5, 10, and 16 days,
respectively (Salby, 1981a, b), due to influences of the vertical and meridional variation
of the mean horizontal winds and temperature.
Among the gravest modes, the quasi-5-day wave (Q5DW) and quasi-16-day
wave (Q16DW) have extensively been studied through observations, modeling, and
assimilation products: Ground-based observations (e.g., Day and Mitchell, 2010; He et
al., 2020; Mitra et al., 2022), satellite observations (e.g., Forbes and Zhang, 2017;
Huang et al., 2022), reanalysis data (e.g., Huang et al., 2017), and simulations (e.g., Qin
et al., 2021). Using meteor radars (MRs) located in the northern and southern polar
regions, Day and Mitchell (2010) showed that PW activity is strong during winter and
the seasonal variation of PW is similar in both polar regions. According to Qin et al.
(2021) and Mitra et al. (2022), the barotropic and baroclinic instabilities are the possible



sources of Q5DW and Q16DW in that the waves can draw energy from the mean flow
in the instability region. The disturbance of zonal-mean flow frequently occurs during
the large-scale meteorological events such as sudden stratospheric warming (SSW). It
has been reported that the amplitude of Q5DW or Q16DW increases during SSW events
(Eswaraiah et al., 2016; Lee et al., 2021; Li et al., 2021; Ma et al., 2022). In addition,
the amplified PWs can modulate the periods of tides through the in-situ nonlinear
interaction, resulting ionospheric disturbances during SSW (e.g., Goncharenko et al.,
2020; Forbes et al., 2021; Liu et al., 2021; Qin et al., 2019).
In contrast, the westward propagating quasi-10-day wave (Q10DW) with zonal
wavenumber 1 (W1) has received little attention compared to the other gravest normal
modes. Forbes and Zhang (2015) showed that Q10DW-W1 has a mean period of $9.8 \pm$
$0.4$ days using the temperature measurements from the Sounding of the Atmosphere
using Broadband Emission Radiometry (SABER) instrument mounted on NASA's
TIMED (Thermosphere Ionosphere Mesosphere Energetics Dynamics) satellite in
2002–2013. They presented that the large amplitude of Q10DW-W1 is found in the
high-latitude mesosphere and lower thermosphere (MLT) region of both hemispheres in
equinoxes, although their results are limited to the latitude of 50° because of the yaw
cycle of the satellite. Hirooka (2000) reported that the global structure of Q10DW-W1
using the Improved Stratosphere and Mesospheric Souder (ISAMS) instrument aboard
Upper Atmosphere Research Satellite (UARS) from November 1991 to May 1992. The
results also showed that the Q10DW-W1 is active during equinoxes and winter at 0.1
hPa (~65 km). In addition, it is found that nonuniform and background zonal wind field
can influence the structure of the wave in the mesosphere. The amplitude of the
Q10DW-W1 is uniform or decays in the vertical near the mesopause, and it does not



increase above the mesosphere, even though the critical layer is absent. Using the
airglow intensities simulated by the global circulation model assimilated by the
reanalysis data from ground to 30 km, Egito et al. (2017) also found that the 10-day
oscillation is dominant from autumn to spring in the mid-latitude MLT region. More
recently, Huang et al. (2021) investigated the Q10DW activity based on the Modern-Era
Retrospective analysis for Research and Applications version 2 (MERRA-2) reanalysis
data. They showed that the dominant components of Q10DW are westward-propagating
wave with zonal wavenumber 1 during winter and spring in the stratosphere and
mesosphere and eastward-propagating waves with zonal wavenumber 1 and 2, which
are excited in the mesospheric instability region. Although both westward and eastward
Q10DW modes are found, they mainly focus on the eastward propagating Q10DW.
Some studies have investigated the climatological and general properties of
Q10DW-W1 activities in the mid- and low-latitudes, but their seasonal variation in the
high-latitude MLT region has not been fully explored. In addition, the amplification
mechanism of Q10DW-W1 still has not been investigated. In the present study, we
focus on the seasonal variation of Q10DW-W1 in the Southern Hemisphere (SH) high-
latitude MLT region using MRs located in Antarctica. Plus, we carry out numerical
simulations using the Specified Dynamics version of the Whole Atmosphere
Community Climate Model (SD-WACCM) nudged towards MERRA-2 reanalysis data
in order to elucidate the observed Q10DW-W1 and its amplification mechanism.
Section 2 describes two MRs located in the Davis station (68.6°S, 77.9°E) and King
Sejong Station (KSS; 62.2°S, 58.8°W) and how we obtain Q10DW-W1 from the
observations. Also, the SD-WACCM experiments and Microwave Limb Sounder
(MLS) data used for validation are described in Section 2. Results are presented in



Section 3. In Section 3.1, we show seasonal variation of observed and modeled
Q10DW-W1 in the SH high-latitude MLT region. The amplification mechanism of
Q10DW is discussed in Section 3.2. Q10DW activities from SD-WACCM simulations
are demonstrated in Section 3.3. In Section 4, the results are summarized, and their
implications are discussed.

**2. Data and Method**
**2.1 Meteor Radars**

In this study, we use two MRs located in the Davis station (68.6°S, 77.9°E) and

King Sejong Station (KSS; 62.2°S, 58.8°W), Antarctica from 2012 to 2016. The
operating frequencies of both Davis and KSS MR are 33.2 MHz and the peak powers
are 6.8 kW and 12 kW, respectively. Details of the operation parameters of Davis and
KSS are summarized in Holdsworth et al. (2008) and Lee et al. (2018), respectively. A
large number of studies has been performed to investigate the PW or tidal activities in
the MLT region with a single-station measurements of horizontal winds from an MR
(e.g., Eswaraiah et al., 2019; Luo et al., 2021; Wang et al., 2021; Liu et al., 2022; Lee et
al., 2021). However, single-station analysis has a limitation in diagnosing the wave
propagation direction, and thus most of such studies focused on the timing of
occurrence and amplitude variations of wave with a particular periodicity. For detailed
analysis of PWs based on the Rossby normal modes, propagation directions and
wavenumbers need to be considered. Recently, He et al. (2018) developed a method of
estimating wave propagation direction and wavenumber as well as amplitude by
adopting Phase Differencing Technique (PDT) to longitudinally separated MR



observations based on the method of Walker et al. (2004). Since the longitude
difference ($\lambda_\Delta$) between Davis and KSS is about 137°, it is appropriate for analyzing
PWs with zonal wavenumber 1 by applying the PDT. In order to estimate the zonal
wavenumber ($s$), we first make a continuous wavelet transform from the daily-mean
Davis and KSS MR data ($W_{(f,t)}^{Davis}, W_{(f,t)}^{KSS}$), respectively, using Morlet wavelet function
as a mother wavelet function (Torrence and Compo, 1998). Then, cross wavelet
spectrum $C_{(f,t)}$ is derived: $C_{(f,t)} = W_{(f,t)}^{*Davis} W_{(f,t)}^{KSS}$, where * denotes the complex
conjugate. Using the phase difference ($\theta_\Delta$) obtained from $\theta_\Delta = \text{Arg}(C_{(f,t)})$ at a given
frequency and time, we estimate zonal wavenumber ($s$): $s = (-\theta_\Delta/(2\pi) + C)/\lambda_\Delta$. In
this study, we focus on the PW activity with $s = 1$, and the number of whole wave cycle
($C$) between two stations is set to be zero (see He et al., 2018 for detailed PDT analysis).
Classical wave theory shows that the latitudinal structures of zonal wind and
meridional wind for Q10DW normal mode from the Laplace tidal equation are
antisymmetric and symmetric with respect to the equator, respectively (e.g., Figure 1 in
Yamazaki and Matthias, 2019). The magnitude of Q10DW-W1 has maxima at the
latitude of 25° and poles for zonal and meridional wind components, respectively.
Around the latitude of 65°S close to the latitudes of the two MR observation sites, the
normalized amplitude of Q10DW-W1 normal mode for the zonal wind is nearly zero,
but the normalized normal mode magnitude for the meridional wind is larger than the
half of the maximum magnitude for the meridional wind (Yamazaki and Matthias,
2019). For this reason, daily-mean meridional wind data from the MRs is used for the
Q10DW analysis.



## 2.2 SD-WACCM


In this study, for detailed analysis of the observed Q10DW-W1 activity and its
amplification mechanism, we compare observational results with Q10DW-W1
simulated using the Specified Dynamics (SD) version of WACCM version 4 (Marsh et
al., 2013). WACCM4 is a high-top (up to the lower thermosphere about 140 km)
atmospheric component model of the Community Earth System Model developed at the
National Center for Atmospheric Research. WACCM4 employs Community
Atmospheric Model (CAM) version 4 physics package. The default horizontal
resolution of WACCM4 is 1.9°×2.5° (lat. × long.), and it uses the 88 hybrid sigma
vertical levels for the SD mode. Since we focus on the PWs such as Q10DW-W1, daily-
mean values from the SD-WACCM are used. In this study, two SD-WACCM
experiments with two different nudging depths (EXP60 and EXP75) are performed. In
the EXP60 and EXP75, model variables are nudged towards the MERRA-2 reanalysis
data from surface to about 60 km in altitude and 75 km, respectively. The MERRA-2
reanalysis is produced by assimilating various types of observations into the Goddard
Earth Observing System version 6 (GEOS6) global model (Gelaro et al., 2017). In
addition to conventional meteorological observations and operational satellite
measurements, the Earth Observing System (EOS) Aura MLS temperature data are
included in the assimilation procedure of the MERRA-2 above 5 hPa (~37 km). As a
result, MERRA-2 reanalysis can reflect the MLT variabilities. As suggested by
Brakebusch et al. (2013), nudging coefficients for EXP60 and EXP75 are 0.01 s$^{-1}$ below
the altitudes of 50 km and 65 km, respectively, and they linearly decrease and become
zero above the altitudes of 60 km and 75 km, respectively.



WACCM simulation requires the data of sea surface temperature, sea ice
fraction, solar and geomagnetic indices, and ionization rate by energetic particle
precipitation (EPP) for the time period of simulations. The sea surface temperature and
sea ice fraction data are produced by the NOAA Optimum Interpolation (Reynolds et
al., 2002). The solar and geomagnetic indices are obtained from NASA GSFC/SPDF
OMNIWeb interface (https://omniweb.gsfc.nasa.gov/ow.html). The EPP ionization rate
is provided by the CCMI reference-C2 data for the period of 1960–2100 (Eyring et al.,
2013). Regarding MLT dynamics, effects of gravity wave drag (GWD) are crucial.
WACCM includes a suite of GWD parameterizations (Richter et al., 2010) for effects of
unresolved GW momentum transfer from orography (McFarlane, 1987), deep
convection (Beres et al., 2005), and frontal activity (Charron and Manzini, 2002). SD-
WACCM simulations start from January 1, 2011 and end at the end of 2016. First one-
year results are discarded as a spin-up, and results for 2012–2016 are compared with
MR observations.

**2.3 MLS**
For validation of Q10DW-W1 estimates obtained from MR observations, we
derive the geostrophic winds from geopotential height (GPH) data (version 5.1 product)
measured using MLS onboard the NASA's EOS Aura satellite (Schwartz et al., 2008).
Geostrophic wind components are computed following Matthias and Ern (2018). The
Aura satellite launched on July 2004 is in a sun-synchronous orbit with an altitude of
705 km. Spatial coverage of MLS instrument is from 82°S to 82°N with a 165 km
resolution along the track. The sun-synchronous orbit of Aura satellite can provide a



global coverage data per day with about 15 orbits. The global coverage of GPH is
produced using daily mean values in 5°×5° (lat. × long.) grids. In this process, GPH
data is filtered on the basis of the recommended precision, status, quality, and
convergence thresholds of Version 5.0 Level 2 and 3 data quality and description
document (https://mls.jpl.nasa.gov/data/v5-0_data_quality_document.pdf).

**3. Results and Discussion**
**3.1 Seasonal variation of Q10DW-W1 in the MLT region**

The perturbation meridional wind for Q10DW-W1 is symmetric in latitude

about the equator as mentioned earlier. Therefore, in order to extract and analyze
Q10DW-W1, it is necessary to confirm whether the latitudinal structure of Q10DW-W1
has the hemispheric symmetry. Although the KSS and Davis MR observations can
provide information about the longitudinal propagation of Q10DW-W1, it is impossible
to estimate the latitudinal structure using these radars alone. In this study, the
meridional geostrophic winds obtained from the MLS geopotential data are used to
confirm the hemispheric symmetry of Q10DW-W1 estimated from MRs. The
amplitudes of Q10DW-W1 in the MLS are obtained using the two-dimensional Fast
Fourier transform (FFT) of the geostrophic meridional winds averaged over the height
range of 80–90 km in time (30-day sliding window) and longitude domain. The time-
latitude cross section of the amplitude of Q10DW-W1 derived from the MLS
geostrophic meridional wind averaged over the height range of 80–90 km is presented
in the Supplement (Fig. S1). Hereafter, the Q10DW denotes westward-propagating
quasi-10-day normal mode wave with zonal wavenumber 1 and the hemispheric



symmetry, where quasi-10-day periodicity means the periods between 9 and 11 days.
Unless the hemispheric symmetry is satisfied, the analyzed westward propagating
signals with zonal wavenumber 1 are referred to as quasi-10-day-like oscillations
(Q10DOs).
Figure 1 shows the time-height distributions of the amplitudes of Q10DWs and
Q10DOs derived from the daily-mean meridional winds observed at the Davis and KSS
MRs using the PDT method. The regions shaded in gray represent the time periods
when the hemispheric symmetry is not found in the MLS results as shown in Fig. S1.
The time periods of the hemispheric symmetries are defined by the periods when the
amplitudes of the MLS meridional geostrophic winds (vertically averaged over 80–90
km) with quasi-10-day periodicity exceed 3.5 m s$^{-1}$ in both 60°N–80°N and 60°S–80°S.
The MLS results in solstices are generally shaded in gray (see Fig. S1). This result
indicates that Q10DWs in a form of normal modes are found during equinoxes, which is
consistent with the results from Forbes and Zhang (2015). Using the periods of the
hemispheric symmetry of the Q10DW obtained from the MLS, we identify the normal
mode Q10DW from the Davis and KSS MR observations.
The 5-yr average (The bottom-most panel of Fig. 1) between 2012 and 2016
indicates that the Q10DWs are generally enhanced from late February to April and from
late August to September in the altitude range of 82–98 km with the maximum
amplitude of 27.2 m s$^{-1}$. The Q10DWs are usually more amplified in early spring from
late August to September with the largest amplitudes around the altitudes of 90–95 km.
Large amplitudes are found in winter (July to mid-August), but they are unlikely to
represent the normal mode Q10DWs, as it is clear from the gray shading in winter.
According to Wang et al. (2021), the nonlinear wave-wave interaction can generate



Q10DOs in southern winter. Their Q10DOs are eastward propagating, interacting with
stationary PWs with zonal wavenumber 1. Meanwhile, the Q10DWs and Q10DOs (Fig.
1) obtained from two MRs using the PDT method are westward propagating.
Understanding of the mechanisms of the winter-time westward-propagating Q10DOs is
beyond the scope of this study, and it requires continuing researches.
For individual years, it is also found that the amplitude of Q10DW is generally
large in equinoxes (see panels for each year in Figs. 1 and S1). During March–April
(autumn), active Q10DWs are identified, and their amplitudes reach up to ~33 m s$^{-1}$ in
2014 and 2015. Particularly, the peak in September (spring) is prominent in 2016. These
MR observation results are remarkably consistent with results obtained using satellite
geopotential height in the SH high-latitude region (Forbes and Zhang, 2015).
Occasionally, large amplitude Q10DWs are observed near the altitude of 98–100 km in
equinoxes (e.g., April 2015), but results around 100 km can be less reliable because the
number of MR echoes above 96 km is much smaller than that around 90 km (Lee et al.,

2022).

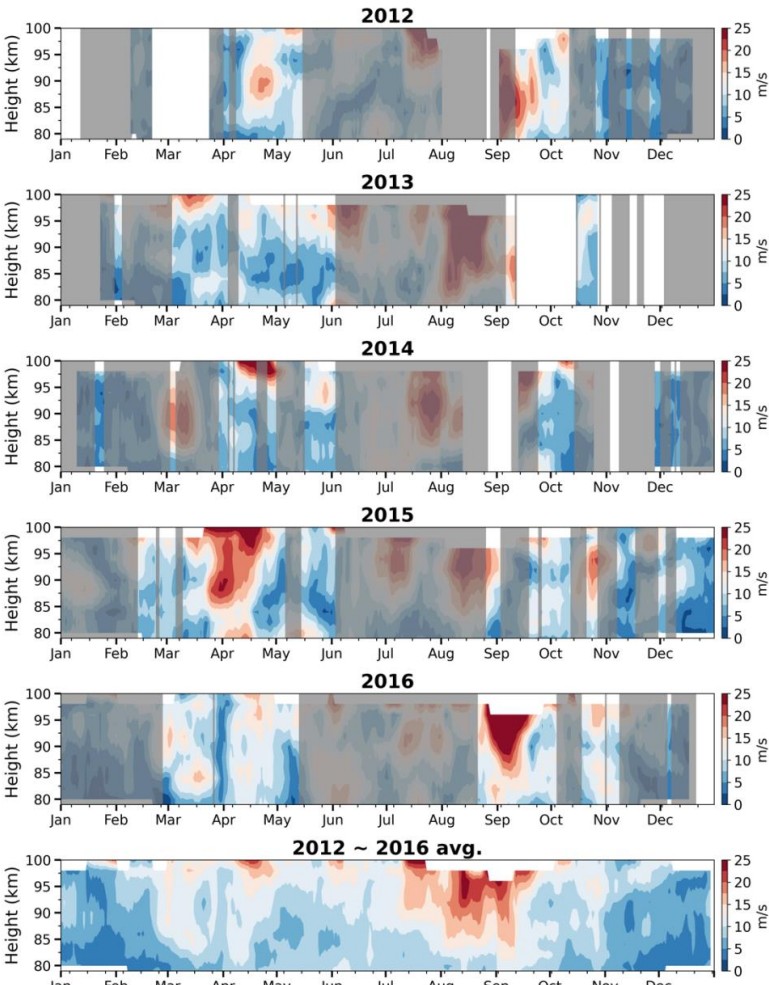


**Figure 1.** Time-height distributions of the amplitudes of Q10DWs (unshaded region)

and Q10DOs (shaded region) derived from meridional winds observed by MRs at Davis

and KSS for 2012–2016. The bottom-most panel shows the 5-yr average from 2012 to

2016. The gray shading represents time periods where the hemispheric symmetry is

unclear in the MLS results.



Figure 2 demonstrates the time-height distributions of the amplitudes of

Q10DWs and Q10DOs around the latitude of 63°S in the EXP75 SD-WACCM
simulation for the altitude range of 60–110 km for 2012–2016, along with the
hemispheric symmetry period obtained from the MLS results. The bottom-most panel of
Fig. 2 shows the 5-yr average from 2012 to 2016. The amplitudes are obtained by
decomposing the meridional winds obtained from the simulation into westward
propagating Fourier modes with zonal wavenumber 1 using the 2D FFT in time (30-day
sliding window) and longitude domain around 63°S. From Fig. 2, it is clear that the
seasonal variations of Q10DW amplitudes obtained from the simulation have year-to-
year variations, as in the Q10DW amplitudes derived from the two MRs. However, the
Q10DW activities observed from the MR observations are generally larger than those in
the EXP75 simulation (see Fig. 1).

The 5-yr average in Fig. 2 shows that there are four main time periods

(February, April, September, November) when the modeled Q10DWs and Q10DOs are
active in the EXP75. The time periods in April and September are consistent with the
MR observations in terms of Q10DW amplitudes and the hemispheric symmetry
obtained from the MLS, but the other periods are not. The active signals simulated in
February and November do not appear to be normal mode Q10DWs because the
hemispheric symmetry is not seen in the MLS data during February and November. For
a more comprehensive understanding of the Q10DOs in the EXP75 during February and
November, we will discuss in more detail later in Section 3.3 by comparing between the
EXP75 and EXP60.

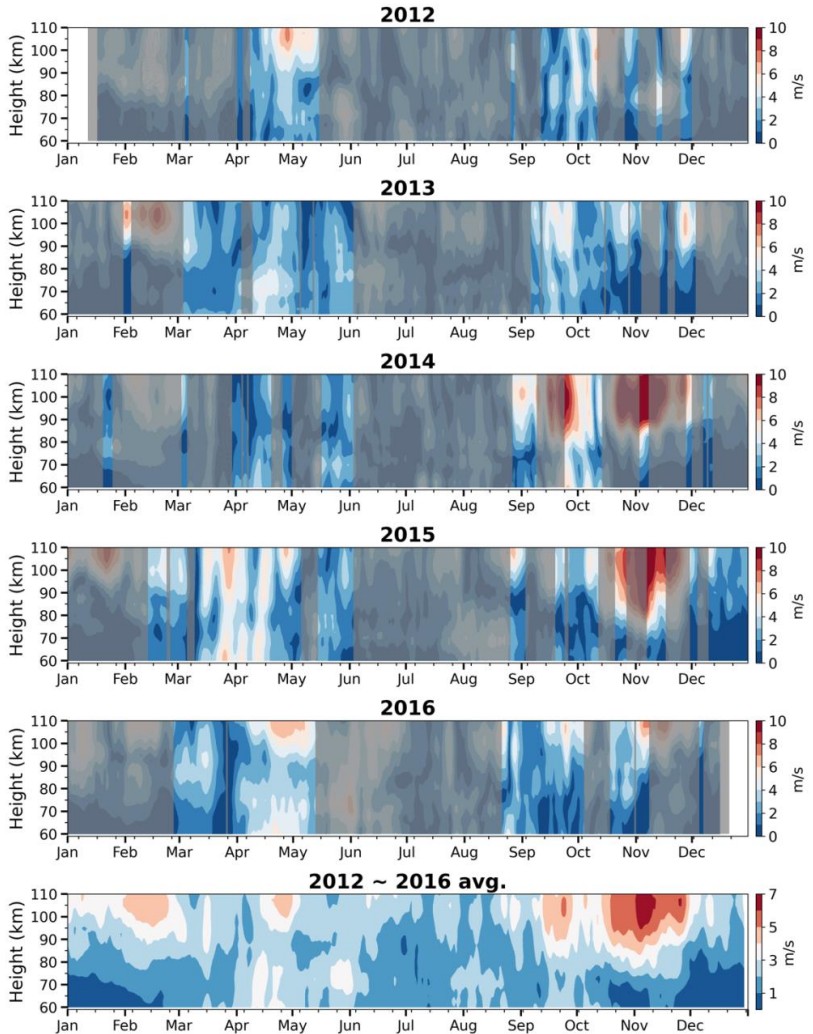

**Figure 2.** Time-height distributions of the amplitudes of Q10DWs (unshaded region) and Q10DOs (shaded region) around 63°S for 2012–2016 in the EXP75. The bottom-most panel shows the 5-yr average between 2012 and 2016. The gray shaded areas represent periods where the hemispheric symmetry is not observed in the MLS results.



Figure 3 shows time series of the normalized amplitudes of Q10DWs and
Q10DOs obtained from the MR observations (black) and EXP75 simulation (blue).
Normalization is carried out by averaging the amplitudes in the altitude range between
80 and 100 km and dividing the 5-yr averaged values by the respective maximum
values in the same altitude range. We select the dates when (i) the amplitudes obtained
from both MRs and EXP75 exceed their respective 5-yr mean values, (ii) their
correlation is relatively large (> 0.6), and (iii) the hemispheric symmetry occurs in the
MLS results. The correlation coefficients are computed for sliding 7-day windows with
1-day step. The dates when the three criteria are satisfied are represented by yellow
boxes on abscissa in Fig. 3. The total number of the dates when the Q10DW was
substantially active in both observations and model (EXP75) is 46. Using EXP75 results
on the selected dates, the amplification mechanisms of the observed Q10DW will be
discussed.

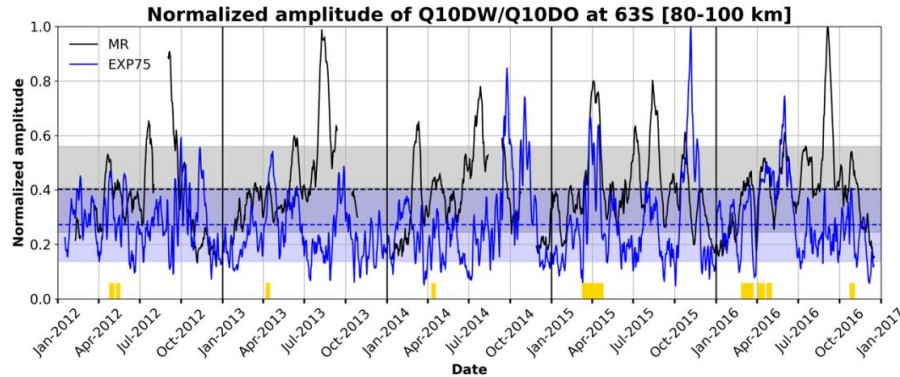

**Figure 3.** Time series of normalized amplitudes of Q10DW/Q10DOs from the

observations (black line) and EXP75 simulation (blue line). The dashed lines and

shaded areas represent the mean and standard deviation of normalized amplitude of

Q10DW/Q10DOs from the observations (black) and EXP75 (blue), respectively.

Yellow boxes on abscissa indicate the dates when the normalized amplitudes from both

MRs and EXP75 can be considered to be those of the normal mode Q10DWs.





**3.2 Amplification mechanisms of Q10DW**


The amplitude of upward propagating PWs grows with height when their
vertical propagation is allowed, but it can decrease with height in the evanescent region
where the square of refractive index $n^2$ becomes negative. Regions of negative $n^2$ are
often accompanied by regions of the negative latitudinal gradient of zonal-mean
potential vorticity ($\bar{q}_\phi$), where $\bar{q}$ is the zonal-mean quasi-geostrophic potential vorticity
(QGPV), the overbar denotes zonal averaging, $\phi$ is the latitude, and the subscript $\phi$
denotes the partial derivative in the latitudinal direction. In the regions of negative $\bar{q}_\phi$,
the barotropic and baroclinic instabilities can occur (Matsuno, 1970), and it is known
that PWs can amplify extracting energy from the mean flow while they pass through the
instability regions (Meyer and Forbes, 1997; Cohen et al., 2013). If PWs somehow
reach their critical lines within an instability region, it is possible for these PWs to
tunnel through the critical lines (Rhodes et al., 2021). In case that the evanescent region
is thin enough, and the PWs can reach their critical lines, it is also possible for the
overreflection to take place, resulting in the amplified PWs and the propagation of the
amplified PWs out of the over-reflection region (Lindzen et al., 1980; Rhodes et al.,

2021).

Another possible way of modulating PWs is their excitation by the
nonconservative GW forcing (Song et al., 2020). Nonconservative GWD forcing
(NCGWD; $Z'$) can generate PWs as it is clearly seen from the perturbation QGPV
equation given in the form of wave action conservation equation (1) when diabatic
forcing is ignored in $Z'$ [see Andrews et al. (1987) and Palmer (1982) for details]:





$$\frac{\partial A}{\partial t} + \nabla \cdot \mathbf{F} = \rho_0 \overline{Z' q'_{(M)}} / (\bar{q}_\phi / \mathrm{a}), \tag{1}$$

where $a$ is the earth's mean radius; $\rho_0$ is the reference density given as an exponentially
decreasing function of log-pressure height $z$; the prime denotes the perturbation from the
respective zonal mean; $A$ is the wave-activity density in the spherical QG system; $q'_{(M)}$ is
the perturbation of modified QGPV, modified to consider the planetary vorticity
advection by the isallobaric meridional wind in spherical geometry (Matsuno, 1970;
Palmer, 1982); $Z'$ is the curl of the horizontal GWD perturbation; $\nabla \cdot \mathbf{F}$ is the divergence
of Eliassen-Palm (EP) flux ($\mathbf{F}$), and the flux $\mathbf{F}$ is considered to be the wave-activity flux
given by $\mathbf{F} = \mathbf{c}_g A$ in the QG framework, where $\mathbf{c}_g$ is the group velocity in the latitude-
height domain.

In (1), the wave-activity density $A$ and the modified QGPV perturbation $q'_{(M)}$ are

given in spherical geometry (Palmer, 1982), respectively, as follows:

$$A = a \cos \phi \, \frac{1}{2} \rho_0 \frac{\overline{q'^2_{(M)}}}{\bar{q}_\phi / a}, \tag{2}$$
$$q'_{(M)} = \frac{v'_\lambda}{a \cos \phi} - \frac{f}{a \cos \phi} \left( \frac{u' \cos \phi}{f} \right)_\phi + \frac{f}{\rho_0} \left( \rho_0 \frac{\theta'}{\bar{\theta}_z} \right)_z, \tag{3}$$

where $u$ and $v$ are zonal and meridional wind components, respectively; $\lambda$ is the
longitude; $f$ is the Coriolis parameter; $\theta$ is the potential temperature. The subscript $\lambda$ and
$z$ mean the partial derivatives in longitude and vertical directions, respectively.



For understanding of amplification of PWs around the instability regions, the
barotropic and baroclinic instability regions are determined by the negative sign of $\bar{q}_\phi$
(Andrews et al. 1987) given by:

$\bar{q}_\phi = 2\Omega \cos\phi - \left[\frac{(\bar{u}\cos\phi)_\phi}{a\cos\phi}\right]_\phi - \frac{a}{\rho_0}\left(\frac{\rho_0 f^2}{N^2}\bar{u}_z\right)_z ,$          (4)

where $\Omega$ is the earth's rotation rate and $N$ is the buoyancy frequency. The negative sign
of $\bar{q}_\phi$ is a necessary condition of the barotropic and baroclinic instabilities. The second
(with negative sign) and third (with negative sign) terms on the right-hand side of (4)
represent the meridional and vertical curvatures of the zonal-mean zonal wind,
respectively. If the second or third term is dominant, $\bar{q}_\phi$ can become negative, and the
instabilities can take place.
The square of refractive index $n^2$ is used to analyze the propagation
characteristics of PWs and depends on the mean QGPV gradient as follows:

$n^2 = \frac{\bar{q}_\phi}{a(\bar{u}-c)} - \frac{s^2}{a^2\cos^2\phi} - \frac{f^2}{4N^2H^2},$          (5)

where $c$ is the zonal phase speed of single PW (i.e., $c = 2\pi a \cos\phi /(s\tau)$; $s$ is the zonal
wavenumber, and $\tau$ is the wave period), and the constant scale height $H$ is set equal to 7
km. The propagation of PWs is possible in regions of positive $n^2$. On the other hand,
PWs can be reflected or be evanescent in the region where $n^2 < 0$ (Matsuno, 1970).





In order to analyze the wave propagation and wave activity for the selected dates
for Q10DWs (or Q10DOs) found in MRs and model simulations, we use the EP flux as
diagnostic tools, derived in the Transformed Eulerian-Mean framework for the spherical
QG system (Palmer, 1982; Andrews et al., 1987). In the spherical geometry, the
meridional ($F^\phi$) and vertical ($F^z$) components of the EP flux $\mathbf{F} \equiv (0, F^\phi, F^z)$ are given
by

$\quad F^\phi = -\rho_0 a \cos\phi \, \overline{u'v'}$ ,      (6)
$\quad F^z = \rho_0 a \cos\phi \, f \, \overline{v'\theta'}/\bar{\theta}_z$ ,      (7)

Figure 4 shows the EP flux $\mathbf{F}$ and wave activity density normalized by $\rho_0 \, a\cos\phi$
for Q10DWs in the EXP75. The propagation inhibition region ($n^2 < 0$) and the
contours of zonal-mean zonal wind are overplotted. Thick green and black lines indicate
the regions of $\bar{q}_\phi = 0$ and of critical lines for Q10DWs, respectively. The critical lines
are plotted by computing the zonal phase speed ($c$) of Q10DW: $c = 2\pi a \cos\phi / (s\tau)$,
where $s = 1$ and $\tau = 10$ day. The wave-activity density is shaded in blue and red
depending on its sign. For the EP flux vector, $\mathbf{F}/\mathrm{sgn}(A) (= \mathbf{c}_g|A|)$, rather than $\mathbf{F}$ itself
($= \mathbf{c}_g A$), is plotted such that the EP flux can always be parallel to the local group
velocity of Q10DWs regardless of the instability regions where $\bar{q}_\phi < 0$ and thus $A < 0$.
For better illustration of the EP flux in the atmosphere where its density decreases
exponentially with height, the meridional and vertical components of EP flux are scaled



by $(p_s/p)^{0.85}[F^\phi/(a\pi), F^z/(3 \times 10^5)]$ (Edmon et al., 1980; Gan et al., 2018), where $p_s$
and $p$ are the surface and atmospheric pressures, respectively.

For Figure 4, we select the four dates of (a) 30 April 2012, (b) 11 April 2013, (c)

6 April 2015, and (d) 29 October 2016 when the three criteria mentioned in Fig. 3 are
satisfied (see yellow boxes in Fig. 3). That is, the normalized amplitudes of Q10DWs
from both MRs and EXP75 are larger than its average, the correlation coefficient is
larger than 0.6, and the hemispheric symmetry is found in the MLS results. The 30
April 2012 case (Fig. 4a) shows that the stratospheric jet is located around (40°S–60°S,
55 km) in the latitude-height domain and that there is a predominant branch of upward
and equatorward Q10DW EP flux vectors across the center of the stratospheric jet. In
the high-latitude mesosphere, there are two regions where both the large-scale
instability ($\bar{q}_\phi < 0$) and evanescence ($n^2 < 0$) take place, and they are located in
(55°S–65°S, 60–85 km) and (65°S–80°S, 70–110 km), respectively. Along the
instability boundaries (green lines), large positive or negative Q10DW activities are
found. Divergent EP flux vectors in the meridional direction are clearly seen around the
instability region located at (53°S, 65–75 km), which implies the excitation of Q10DWs
in association with the instability. In the region of MR observations (60°S–65°S, 85–
100 km), substantially amplified Q10DW activity appears, and the equatorward
Q10DW EP flux towards the MR sites is found over the amplified Q10DW activity.

Figures 4b demonstrates the case of 11 April 2013. One major branch of

Q10DW EP flux vectors (Fig. 4b) originate from the stratospheric jet located at (55°S–
60°S, 45–60 km). In the southern and upper side of the stratospheric jet, the instability
and evanescent region extends from 45 km to 70 km height in the latitude of 50°S–



75°S. Above the instability region, distinct region of strong wave activity is found
around (50°S–65°S, 65–90 km), and this region is partially overlapped by the MR
observation region. Around this region, the Q10DW EP flux is directed downward and
poleward inside of the instability region (within green line). The Q10DW EP flux is
directed upward and equatorward outside and above the instability region. This
diverging pattern of EP flux around the instability region also shows the possibility of
the excitation of Q10DW in association with the instability.

For 6 April 2015 case (Fig. 4c), the structure of wave-activity density and

instability regions are similar to the 30 April 2012 case (Fig. 4a). The instability and
evanescent regions occur around (60°S–80°S, 70–100 km). Along the instability
boundaries, there are strong positive and negative wave-activity densities, and this
region of strong wave activities includes the MR observation region. Again, the
divergent of Q10DW fluxes appears in the upper part of the instability region around
(60°S–70°S, 80–100 km). The Q10DW propagates upward and equatorward outside of
the instability region and downward inside of the instability region, as in the other dates
shown in Figs. 4a and 4b.

In 29 October 2016 case (Figure 4d), the center of stratospheric jet is located

around (60°S–70°S, 20–30 km). Above the stratospheric jet, the eastward wind turns
westward around the altitude of 60 km. Within the region of westward wind, the
instability and evanescent regions are found. In addition, the critical lines exist inside
the instability region. The overreflection or transmission process can take place near the
critical lines as we mentioned. Notably, the significantly large positive and negative
wave-activity density regions are found around (45°S–70°S, 60–90 km) near the





instability boundaries, and these regions are partially overlapped by the MR observation
region. This result suggests that the observed amplification of Q10DW may be
attributed to the overreflection or transmitted process. The EP flux of Q10DW
predominantly propagates upward and equatorward away from the strong wave-activity
region around (60°S, 60–70 km) with weak poleward propagation of Q10DW towards
the instability region across the critical lines.

For all the cases shown in Fig. 4, the results indicate that a distinct strong wave-

activity density region is located within the area observed by the MRs (around 60°S–
70°S and 80–100 km in height) associated with the large-scale instability region.
Considering the wave-activity density $A$ is directly proportional and inverse
proportional to the $\overline{q'^2}$ and $\bar{q}_\phi$, respectively, it can be thought that the small $\bar{q}_\phi$
contributes the large magnitude of $A$ near the instability region. However, we confirm
that the large $\overline{q'^2}$ is located around the instability region, leading to the overall large
wave-activity density (not shown in here). In addition, the group velocity of the wave is
given by $\mathbf{c}_g = \mathbf{F}/A$. For the selected cases (Fig. 4), the EP flux $\mathbf{F}$ in the MR observation
region is relatively small, while the magnitude of $A$ is comparatively large. This
suggests a small group velocity in this region. These results agree with the study of
Thorncroft et al. (1993), which states that during the amplification of baroclinic waves,
the group velocity tends to be small.

As previously mentioned, Song et al. (2020) proposed that the NCGWD can

generate PWs. In this regard, the resolved GWs ($s \geq 20$) could also play a role in
generating Q10DW. To verify the contribution of NCGWD, we analyze linearized
disturbance QGPV equation (Andrews et al., 1987) for the 4 cases shown in Fig. 4. Our



analysis shows that the contribution of both NCGWD and resolved GW for the Q10DW
is negligible in the MLT region (see Fig. S3 in the Supplement).

These results indicate that the large amplitudes of Q10DW observed in the SH

high-latitude region by the Davis and KSS MRs can originate from the high-latitude
stratosphere-mesosphere region, where the barotropic/baroclinic instability or
overreflection near the critical layer occur.

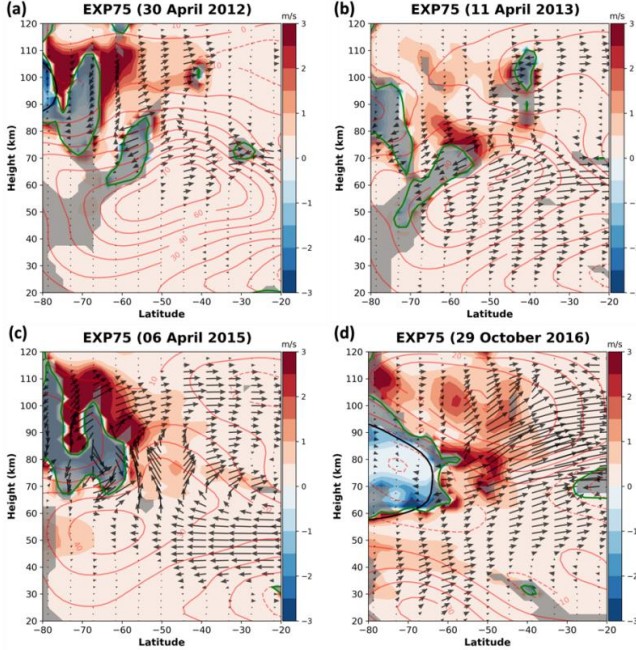


**Figure 4.** EP flux parallel to local group velocity [$\mathbf{F}/\mathrm{sgn}(A)$] and normalized wave

activity density [$A\,(\rho_0 a \cos \phi)^{-1}$ given in the unit of m s$^{-1}$] for the Q10DWs in the

EXP75 on (a) 30 April 2012, (b) 11 April 2013, (c) 6 April 2015, and (d) 29 October

2016. The activity density $A$ is shaded in blue and red depending on its sign. The

boundaries of the instability regions ($\bar{q}_\phi = 0$, green lines), the negative $n^2$ regions (grey





shading), and the red contours for zonal-mean zonal wind are overplotted. For eastward
(westward) zonal-mean zonal wind, contours are plotted in solid (dashed) lines, and
contour interval is 10 m s$^{-1}$.

**3.3 Comparison of Q10DO between SD-WACCM simulations**

This section compares the Q10DOs around the mesospheric instability regions
in the two SD-WACCM simulations (EXP75 and EXP60) for February and November.
February and November are chosen because the amplitudes of modeled Q10DOs are
substantial. The magnitude of Q10DO in the EXP75 is generally smaller than that in the
EXP60, which is more comparable to the MR and MLS observations in which both
Q10DWs and Q10DOs are weak (see Figs. S1 and S2 in the Supplement). Note that
more realistic meteorological fields are nudged throughout the mesosphere in the
EXP75. In this section, comparison between EXP75 and EXP60 for February and
November is carried out to reveal mechanisms behind weak Q10DOs in the EXP75.
Figure 5 demonstrates the properties of Q10DO and background atmospheric
conditions (as shown in Figure 4) for 5 February 2013 and 16 November 2016 when the
Q10DO activity is found to be large in both simulations. The left and right panels of
Fig. 5 are the results from the EXP75 and EXP60, respectively. In Fig. 5, it is clear that
the strong wave-activity density for Q10DO arise in polar regions above the altitude of
70 km in the EXP60, and the magnitude of the EP fluxes in the EXP60 is stronger than
that in EXP75. In addition, in 5 February 2013 for the EXP60 (Fig. 5b), a substantially
strong wave-activity density region is located in the mid-latitude mesospheric region as
well. Around the strong wave-activity regions in the polar upper mesosphere, it is seen
that the EP fluxes of Q10DWs are divergent. In addition, the distinct wave-activity





density of Q10DO regions in the EXP60 occur along the instability regions and critical
lines around (50°S–70°S, 70–110 km) and (20°S–40°S, 65–80 km). On the other hand,
the wave-activity density of Q10DO in the EXP75 (Fig. 5a and 5c) is located at
relatively higher altitudes (80–100 km), and the strength of Q10DO EP flux and wave-
activity density are weaker than EXP60. Moreover, the negative EP flux divergence
(EPFD) is much larger in the EXP60 than in the EXP75 above the altitude of 80 km (not
shown in here).
Our analysis reveals that the larger wave-activity density and EP fluxes in the
EXP60 along the large-scale instability region in the polar upper mesosphere compared
to the EXP75. This indicates that the stronger large-scale instability in the EXP60 can
amplify Q10DO activities, which is consistent with the analysis result that the
barotropic and baroclinic instabilities can be the major sources of the amplification of
traveling PWs (Harvey et al., 2019).

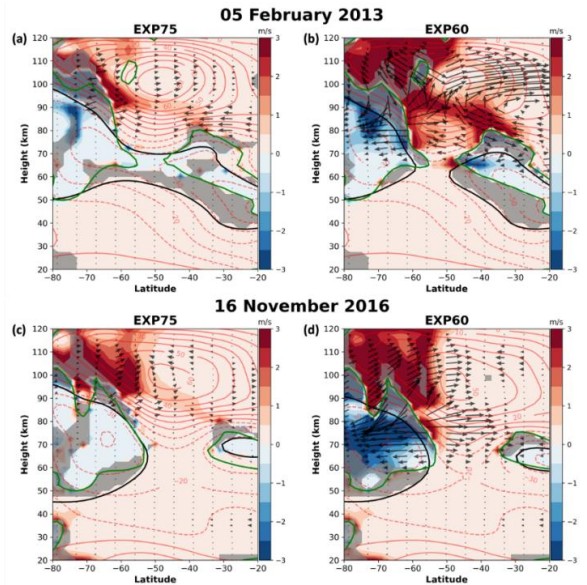




**Figure 5.** Same as Fig. 4 but for (a and b) 5 February 2013 and (c and d) 15 November
2016. The left and right columns represent the results from EXP75 and EXP60,
respectively.

Figure 6 shows the $\bar{q}_\phi$ (normalized by $\Omega$) for 5 February 2013 and 16 November

2016 from the EXP75 (blue), EXP60 (orange), and MLS (green). The normalization
makes $\bar{q}_\phi$ dimensionless. The $\bar{q}_\phi/\Omega$ from MLS is derived in the quasi-geostrophic
framework (Andrews et al., 1987) and it is included as a reference for validation. The
$\bar{q}_\phi/\Omega$ is averaged between the latitudes of 65°S-80°S where the wave-activity density is
strong and large negative $\bar{q}_\phi$ is found in Fig. 5. It is seen that the vertical profiles of
$\bar{q}_\phi/\Omega$ from the EXP75 and MLS have somewhat small negative values and they are
generally similar below the altitude of 75 km, although the difference gradually increase
above the altitude of 75 km. On the other hand, large discrepancies are shown between
EXP75 and EXP60 in the altitudes between 60–80 km. In the EXP60, $\bar{q}_\phi/\Omega$ has much
larger negative values, which suggest the relatively stronger barotropic or baroclinic
instability in the EXP60 and the amplification of the Q10DO in the mid-to-upper
mesosphere in association with the stronger instability in the EXP60.



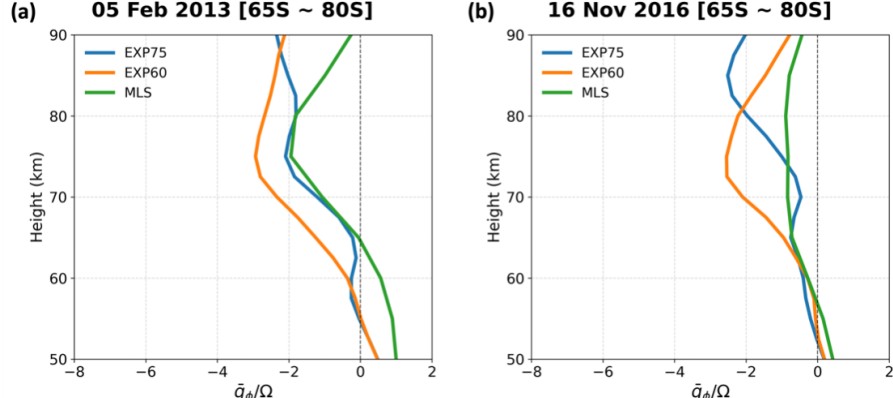


**Figure 6.** $\bar{q}_\phi$ (normalized by $\Omega$) averaged over 65°S–85°S for (a) 5 February 2013 and

(b) 16 November 2016 from the EXP75 (blue), EXP60 (orange), and MLS (green).

The negative $\bar{q}_\phi$ can be induced by latitudinal and vertical curvatures of zonal-

mean zonal wind that correspond to the second and third terms (with negative signs) in

the right side of (4), respectively. Figure 7 shows the second (top panels) and third

(bottom panels) terms, respectively, for 5 February 2013. The difference shown in Figs.

7c and 7f indicate that the larger negative $\bar{q}_\phi$ is located in the lower altitudes in the

EXP60 than in EXP75, inducing the larger instability at 65–75 km in height around

70°S–80°S in the EXP60, which is consistent with Fig. 6. Note that the positive

differences seen at about 65–75 km in the high-latitude regions in Figs. 7c and 7f mean

the larger negative $\bar{q}_\phi$ in the EXP60. Also, it is clear that both vertical and horizontal

shear contribute the stronger barotropic/baroclinic instability in the EXP60 in the mid-

to-upper mesosphere, as shown in Figs. 7a-b and 7d-e. This analysis demonstrates the

mesospheric dynamics specified by the MERRA-2 data up to the altitude of 75 km

reduces the large-scale instability in the mid-to-upper mesosphere in the EXP75. This is



consistent with Sassi et al. (2021) proposed the absence of specification of middle
atmosphere dynamics induce the instability in summer mesospheric westward jet,
leading large traveling PWs.

The wind structure in the MLT region is mainly driven by momentum

deposition from PWs and GWs. Harvey et al. (2019) reported that GWs can change
significantly the vertical shears, leading enhanced instability and larger traveling PWs in
the mesospheric region based on the satellite observations and SD-WACCM
simulations. In addition, the unresolved GW forcing is one of the main factors to
maintain the necessary conditions of barotropic/baroclinic instability in the modeled
mesosphere (Sato et al., 2018). Therefore, in order to better understand the mechanisms
underlying the discrepancies in zonal wind fields and the resulting instability in the
model, it is important to examine the contribution of resolved wave forcing (EPFD) and
GWD forcing on the zonal wind structure in the mesosphere.

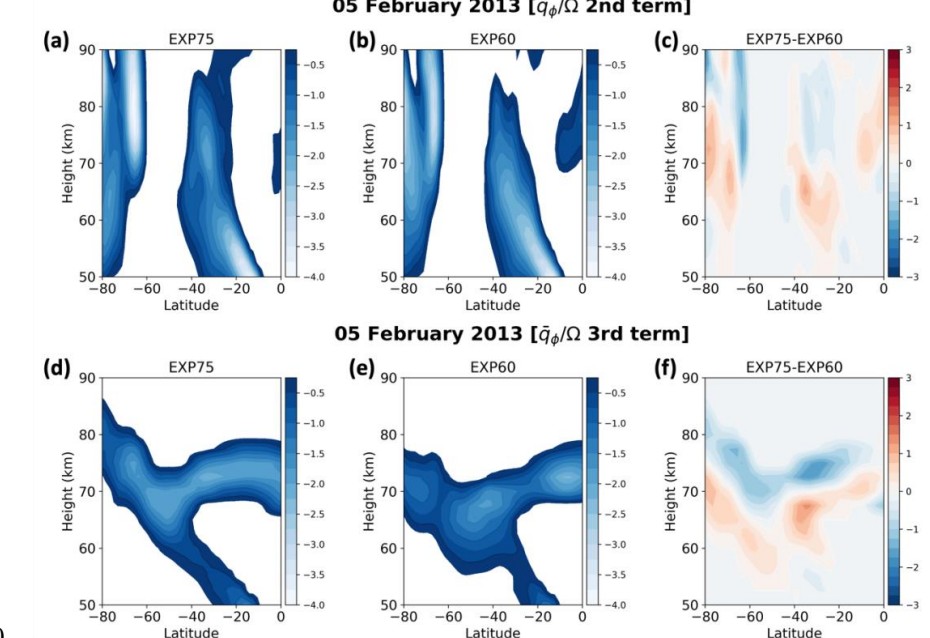

**Figure 7.** Contribution of (top) the meridional variation of the zonally-averaged mean

flow and (bottom) its vertical variation in the instability condition (negative $\bar{q}_\phi$) shown

in (2), respectively, for 5 February 2013. Each column presents the results from (a and

d) the EXP75, (b and e) the EXP60, and (c and f) difference between EXP75 and

EXP60. Only negative values are plotted except for two panels for difference.

Figure 8 shows the latitude-height distributions of zonal-mean zonal wind, zonal

component of GWD and resolved wave forcing (EPFD) in 5 February 2013 for the

EXP75, the EXP60, and the difference between EXP75 and EXP60 (EXP75–EXP60).

The zonal-mean zonal wind, zonal component of GWD, and resolved wave forcing

(EPFD) are calculated through the 21-day averaging (central date ± 10 days). For GWD,

the orographic and nonorographic values are added. In Figs. 8a–b, zero-wind lines are

located around 80 km height in the SH mid-latitude region, indicating the reversal of the



zonal-mean zonal wind due to the eastward momentum forcing from the GWs and
resolved waves. It is clear that the zero-wind line in the EXP60 is located at lower
altitudes by about 5 km compared to the EXP75, which means that eastward GWD and
eastward EPFD from the EXP60 can be larger below the altitude of ~80 km than that
from EXP75. Indeed, the difference field between EXP75 and EXP60 for GWD (Fig.
8f) shows that the eastward GWD from the EXP60 is larger around (60°S, 70 km) than
that from EXP75 as indicated by the negative difference field in those regions. In
addition, the resolved wave forcing (EP flux divergence) is more eastward above the
altitude of 70 km in the mid-to-high latitude regions in the EXP60 than in the EXP75.
This result indicates that the eastward resolved wave forcing also contributes more in
the mid-to-upper mesosphere in the EXP60, resulting in the zonal-mean zonal wind
reversal (westward to eastward wind) in the lower altitude in the EXP60.
As mentioned before, the amplification or modulation of westward-propagating
PWs with zonal wavenumber 1 and a quasi-10-day period due to NCGWD and resolved
GW is negligible (Fig. S3 in Supplement), indicating that the amplification of Q10DW
or Q10DO is mainly related to the baroclinic/barotropic instability. The stronger
instability in the EXP60 around the altitude of 70 km indicates that WACCM simulates
a large meridional and vertical variation of zonal winds compared to the observations in
the mid-to-upper mesosphere, which is likely due to the stronger eastward GWD and
eastward EPFD forcing near 70 km altitude in the EXP60, as shown in Fig. 8. Cohen et
al. (2013) reported that parameterized GWs can generate instability that can generate
resolved waves of which forcing (i.e., EPFD) can compensate GWD. Our results also
show that the increased eastward GWD at 70 km altitude generates instability and it
leads more Q10DO. The EPFD in the EXP60 gives the more eastward forcing above 70





km enhancing the wind reversal in the mid-to-high latitudes, but comparison of Figs. 8f
and 8i indicates that the compensation between GWD and EPFD is roughly valid with
slight shift in the vertical direction. Raising the nudging altitude of MERRA-2
reanalysis data to 75 km from 60 km reduces the instability in the mid-to-upper
mesosphere, leading to decreased the Q10DO activity in the EXP75. Therefore, we
suggest that strong eastward GWD in the mid-to-upper mesosphere in summer need to
be alleviated, which can generate more instability in the SH high-latitude mesosphere
region that can lead to discrepancy from observations.

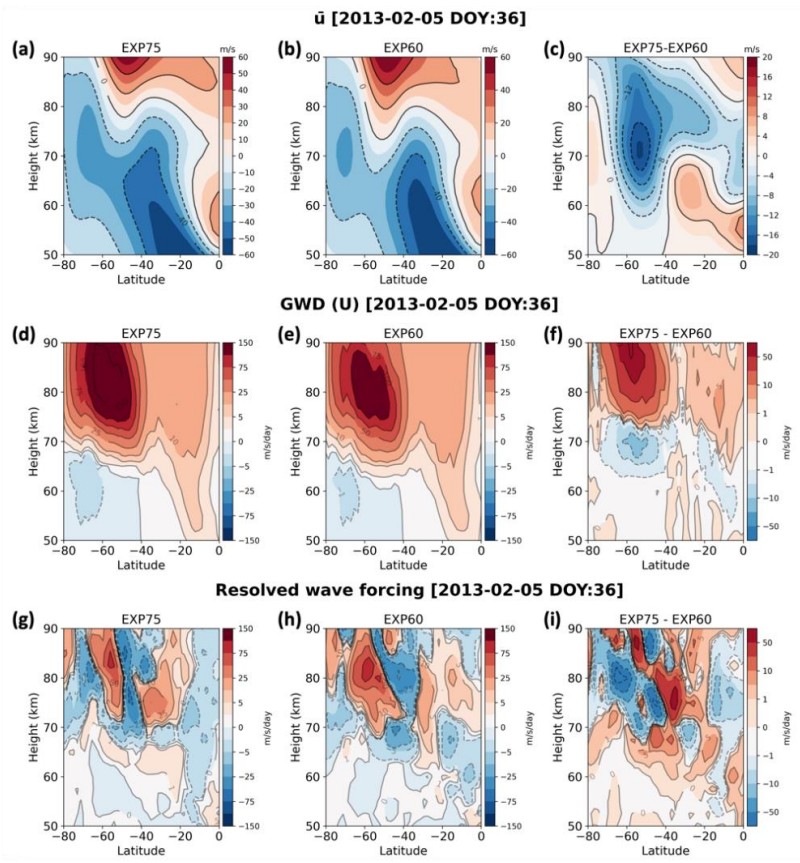

**Figure 8.** Latitude-height distributions of (a–c) zonal-mean zonal wind, (d–f) zonal component of GWD and (g–i) resolved wave forcing (EP flux divergence) in 5 February 2013 for (left) the EXP75, (middle) the EXP60, and (right) difference between EXP75 and EXP60 (EXP75–EXP60).



## 4. Summary

In this paper, the seasonal variation and the amplification mechanism of Q10DW during 2012–2016 in the SH high-latitude regions are investigated using two MRs located in Antarctica, and SD-WACCM simulations.

1. Using the phase difference of meridional winds measured by two MRs, we extract westward-propagating Q10DW with zonal wavenumber 1. The seasonal variation of the observed Q10DW shows that the amplitude is strong during equinoxes, which is consistent with previous studies.

2. In order to elucidate the amplification mechanism of Q10DW observed by MRs during equinoxes, two SD-WACCM experiments are carried out using the MERRA-2 reanalysis data from surface to ~60 km (EXP60) and ~75 km (EXP75), respectively.

3. The temporal variation of the averaged amplitude of Q10DW in the EXP75 during 2012–2016 is in better agreement with the MR observations. Meanwhile, the amplitude of Q10DW in the EXP60 is excessive compared with the observations.

4. Based on the analysis of meridional gradient of the QGPV and wave-activity density, the Q10DW observed in the SH high-latitude region by the MRs originated in situ around the high-latitude stratosphere-mesosphere region, where the large-scale instability or overreflection near the critical lines occur.

5. The unrealistically large magnitude of Q10DO (quasi-10-day-like oscillations without satisfying the hemispheric symmetry unlike Q10DW) is simulated in the EXP60 during February and November. In order to reveal mechanisms of



the large amplitude of Q10DO in the EXP60 during the SH summer, we
compare the meridional gradient of QGPV from EXP75 and EXP60.
6. The results show that specified dynamics with MERRA-2 reanalysis data
mitigate the meridional and vertical variation of zonal winds in the polar mid-
to-upper mesosphere in the EXP75, leading reduction in the large-scale
instability. On the other hand, the large amplitude of Q10DO in the EXP60 is
attributed to the large-scale instability related to the GWD and partially to the
EPFD in the polar mid-to-upper mesosphere. The polar mesospheric GWD can
lead to a strong large-scale instability in the SH high-latitude mesosphere and
unrealistically large amplitude of Q10DO in summer.
The present study on the amplification mechanism of Q10DW during equinoxes, and
the unrealistic Q10DO amplitude in summer provide potential importance of large-scale
instability, which can be to a substantial degree caused by parameterized GWD, during
summer in the polar mesosphere for numerical models. In this paper, we focus on the
Q10DW relating to the large-scale instability and polar mesospheric GWD, but other
normal modes of PW will be considered for future studies.
**Code and Data availability**
The source code of Community Earth System Model 2 (CESM2) developed at
the National Center for Atmospheric Research (NCAR) is available at
https://www.cesm.ucar.edu/models/cesm2. The atmospheric forcing data for specified
dynamics are available from NCAR Research Data Archive (RDA) at
https://rda.ucar.edu.



The Davis station meteor radar data are available from the Australian Antarctic
Data Centre at https://data.aad.gov.au/metadata/records/Davis_33MHz_Meteor_Radar.
The King Sejong Station meteor radar data are available from Korea Polar Data Center
(KPDC) at https://kpdc.kopri.re.kr. The GPH data from the MLS onboard the NASA's
EOS Aura satellite are available from Goddard Earth Science Data and Information
Services Center (GES DISC) at https://daac.gsfc.nasa.gov.
**Author contributions**
WL, ISS, and YHK designed the study. WL and ISS carried out the SD-
WACCM experiments and analysis the observational data. WL wrote the manuscript.
ISS and BGS aided in interpreting the results and worked on the manuscript. All authors
discussed the results and contributed to the final manuscript.
**Competing interests**
The authors declare that they have no conflict of interest.
**Acknowledgements**
This research was supported by the Korea Astronomy and Space Science
Institute under the R&D program (Project No. 2023-1-850-07) supervised by the
Ministry of Science and ICT.



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
