# Peer review of "Quasi-10-day wave activity in the southern high-latitude MLT"

_EGUsphere, 2023_

## Community Comment (CC1)

**Response to Reviewer #1's Comments**

We appreciate your time and effort you have invested to review our manuscript entitled "Quasi-10-day wave activity in the southern high-latitude MLT region and its relation to the large-scale instability and gravity wave drag". Your comments and suggestions give us a valuable opportunity to enhance our manuscript.

We have carefully considered each of your comments and will have revisions to address the concerns raised. Below, we provide detailed responses to your major and minor comments, outlining the modifications we will make to the manuscript. We believe that these changes will be able to significantly improve the quality and clarity of our manuscript.

**General comments:**

**Q. This study examines the generation and propagation of 10-day waves in the southern hemisphere upper atmosphere, using observations from meteor radar winds at two Antarctic stations, with supporting data from SD-WACCM simulations and MLS data. The main conclusion of this study is the identification of regions of barotropic/baroclinic instability in the upper mesosphere and lower thermosphere that can generate such oscillations**

A: The identification of regions of barotropic/baroclinic instability in the upper mesosphere and lower thermosphere as potential generators of oscillations is one of conclusions of our work. However, we would like to emphasize that our study also presents two additional key findings.

1. The first one is the climatology and seasonal variability of the quasi-10-day wave (Q10DW) observed in the Antarctica MLT region. The seasonal variation we present in our manuscript adds to the list of observational reports of mesosphere dynamics that can be used for the validation of model results as well as other observations.

2. The second one is the implication of comparison between EXP60 and EXP75. This comparison suggests that large-scale instabilities due to excessive gravity wave drag (GWD) in the summer mesosphere can make travelling quasi-10-day planetary waves (PWs) overestimated. This finding suggests that the

more accurate representation of GWD is required to improve the mesosphere dynamics in high-top models.

**Major comments**

**Q1. The main conclusion of the study is not new. The authors have ignored existing literature which offers corroborating or alternative investigations: doi 10.1002/grl.50373, 1016/j.asr.2022.10.054, 10.1029/2019JD031599 and 10.1016/j.jastp.2014.06.009, and many others published since Chandran's study.**

A1: We agree that the studies you mention should be incorporated into our manuscript to provide a comprehensive background, and thus we will add these references to the introduction and discussion sections of our revised manuscript.

However, those studies particularly focus on the variation of Q10DW activity during sudden stratospheric warming (SSW) events. Our study describes the 5-year and seasonal variations of Q10DW over the Antarctic region observed using ground-based meteor radars (MRs) rather than Q10DWs during particular events. As mentioned in the introduction, Forbes & Zhang (2015) reported climatology and seasonal variations of Q10DW activity within +-50 degrees latitude in the altitude range of 20–100 km based on the SABER data for 2002–2013 . Our research also contributes to the climatology and seasonal variations at higher latitudes over the Antarctica for z= 80–100 km, Furthermore, our additional conclusion suggests that large-scale instabilities due to strong eastward GWD in the summer mesosphere can lead to the generation of Q10DW as we mentioned above.

**Q2. MERRA-2 is scientifically valid up to 60 km, the uppermost altitude where MLS data is used. Above 60 km and all the way to the lid (~75 km), it is just a sponge layer. Thus, I cannot associate any usefulness to the EXP75 presented in this study, where SD-WACCM is nudged with MERRA-2 fields all the way to MERRA-2 lid: it is like nudging a model with another model. Certainly, EXP60 is more useful but it is discussed only in terms of a difference from the EXP75 and briefly. If the goal of doing this study was to compare two simulations with higher and lower nudging fields, the authors should then be aware of these extant studies: doi 10.1002/2017JD027782, or 10.1002/2015GL065838.**

A2: Since late 2004, the data assimilation system at NASA GSFC GMAO have incorporated the MLS temperature and ozone from 5 hPa up to 0.02 hPa and from 250 hPa to 0.1 hPa, respectively, for the production of the MERRA-2 data (Gelaro et al., 2017; McCormack et al..2021). We think the assimilation of MLS data gives the mesospheric meteorological analysis with a reasonable quality when compared to other mesospheric reanalysis. McCormack et al. (2021) showed the reasonable quality of the mesospheric part of the MERRA-2 data through the intercomparison of the mesospheric assimilation such as the MERRA-2 and results obtained using the higher-top data assimilation systems. The realism of the MERRA-2 in the upper mesosphere is also supported by the personal communication with Drs. Lawrence Coy and Krzysztof Wargan at NASA GSFC GMAO (8 November, 2023). They told us that there is no evidence that the divergence damping layer (Fujiwara et al., 2017) near the top boundary of the GEOS6 model is unduly interfering with the MERRA-2 results. They also said that while there is a lot of variability in the different (mesospheric) assimilation systems, meaning a lot of uncertainty, MERRA-2 seems to fit well in the mix. In fact, Figure 11.42 in Chapter 11 of the Stratosphere-troposphere Processes And their Role in Climate (SPARC) Reanalysis Intercomparison Project (S-RIP) Final Report (Harvey et al., 2021; https://www.sparc-climate.org/sparc-report-no-10) demonstrates that upper mesospheric temperature in the MERRA-2 compares well with the MLS temperature. As reviewer pointed out, there is the sponge layer near the top boundary of the GEOS6 model, but the use of the sponge does based on the divergence damping in the MERRA-2 not seem to degrade seriously the reliability of the MERRA-2 in the mesosphere. A lot of damping mechanisms (Rayleigh friction, 2nd-order hyperdiffusion, and so on) can be used for the sponge layer (Jablonowski and Williamson, 2011). Rayleigh damping and low-order hyperdiffusion can damp significantly model forecast fields, but the divergence damping is often used to effectively and selectively remove high-frequency (noisy) gravity waves keeping the large-scale circulation and planetary wave structure less changed. Below, we compare the horizontal temperature distribution between MLS and MERRA-2 data. The results demonstrate generally good agreement between the MERRA-2 and MLS above 0.1 hPa (Figure R1), suggesting that the EXP75 experiment is not simply nudged towards model forecasts but towards assimilation based on MLS. We will revise our manuscript to include a more detailed justification of this specified dynamics experiments, addressing potential concerns about the validity of simulation results in the upper mesosphere.

[Figure]

**Figure R1** Longitude-Latitude distributions of temperature at three different pressure levels of 0.03 hP a, 0.04 hPa, and 0.07 hPa. For each pressure level, the temperature distributions from (left) MLS and (right) MERRA-2 are presented.

**Q3. MLS data is used in the data assimilation of MERRA-2. I find it odd to attempt to validate the model results that are nudged to MERRA-2 with a dataset that is part of the data assimilation cycle. In other words, an independent data set ought to be used. Why not SABER?**

A3: We agree with your opinion. Independent datasets need to be employed for validation. First of all, however, we will check the availability of SABER data in the Antarctic region. Due to the yaw orbit of TIMES satellite, we need to confirm whether SABER data are available in the high-latitude regions for the specific dates and locations of interest. If the SABER observation is available in the Antarctic areas, we will incorporate the analysis with

SABER into our manuscript to strengthen of our results.

**Q4. The authors identify the occurrence of a 10-day normal mode from the symmetric behavior of MLS geostrophic winds. That is not sufficient: the amplitude structure has nodes, and the phase is expected to be in a specific configuration in order to be 10-day free oscillation. See: DOI: 10.1175/JAS-D-11-0103.1, 10.1111/j.1600-0870.2007.00257.x, and the Salby's papers already cited.**

A4: We agree that a rigorous analysis based on Legendre polynomials and Fast Fourier Transform, which provide latitudinal and longitudinal structures of the normal mode, is the optimal method for identifying the theoretical normal modes. However, It is noteworthy that several studies have recognized disturbance with a quasi-10-day period or other periods of gravest normal modes, westward propagation, and zonal wavenumber 1 as normal mode waves in most of the middle atmosphere (e.g., Forbes and Zhang, 2015; Yamazaki and Matthias, 2019). Forbes and Zhang (2015) observed that the normal mode of Q10DW tends to be contaminated above 80 km altitude, the layer on which our study focuses. In their study, the amplitude and latitudinal structure of Q10DW above z = 80 km are modified by the asymmetry of mean winds between the hemispheres and meridional temperature gradient. In addition, they suggested that the dissipation of gravity wave filtered by the Q10DW wind field can generate a secondary Q10DW by momentum deposition. Our supplementary figure (Fig. S3) demonstrates that this latter factor is unlikely to be a significant contributor to the generation of Q10DW in our cases. In our manuscript, we tried to distinguish Q10DW with normal mode-like structure from oscillations that seem to never have the normal mode-like structure using MLS data. As the reviewer pointed out, however, this distinction may potentially cause confusion, and it will be excluded in our revised manuscript. In the light of removing potential confusions, we will carefully define our terminology in the revised manuscript. The Q10DW will be described as "a westward-propagating (travelling) planetary wave with zonal wavenumber 1, potentially related to the Rossby normal mode". We will incorporate the considerations mentioned here to clarify our analysis and conclusions.

References:

- Jablonowski, C., and Williamson, D. L.: The pros and cons of diffusion, filters and fixers in atmospheric general circulation models, in Numerical Techniques for Global Atmospheric Models, edited by: Lauritzen, P. H., Springer-Verlag, Berlin, Heidelberg, Germany, 381-493, http://doi.org/10.1007/978-

3-642-11640-7_13, 2011.

- McCormack, J. P., Harvey, V. L., Randall, C. E., Pedatella, N., Koshin, D., Sato, K., Coy, L., Watanabe, S., Sassi, F., and Holt, L. A.: Intercomparison of middle atmospheric meteorological analyses for the Northern Hemisphere winter 2009–2010, Atmos. Chem. Phys., 21, 17577–17605, https://doi.org/10.5194/acp-21-17577-2021, 2021.

- Forbes, J. M. and Zhang, X.: Quasi-10-day wave in the atmosphere, J. Geophys. Res.-Atmos., 120, 11,079–11,089, https://doi.org/10.1002/2015jd023327, 2015.

- Fujiwara, M., Wright, J. S., Manney, G. L., Gray, L. J., Anstey, J., Birner, T., Davis, S., Gerber, E. P., Harvey, V. L., Hegglin, M. I., Homeyer, C. R., Knox, J. A., Krüger, K., Lambert, A., Long, C. S., Martineau, P., Molod, A., Monge-Sanz, B. M., Santee, M. L., Tegtmeier, S., Chabrillat, S., Tan, D. G. H., Jackson, D. R., Polavarapu, S., Compo, G. P., Dragani, R., Ebisuzaki, W., Harada, Y., Kobayashi, C., McCarty, W., Onogi, K., Pawson, S., Simmons, A., Wargan, K., Whitaker, J. S., and Zou, C.-Z.: Introduction to the SPARC Reanalysis Intercomparison Project (S-RIP) and overview of the reanalysis systems, Atmos. Chem. Phys., 17, 1417–1452, https://doi.org/10.5194/acp-17-1417-2017, 2017.

- Gelaro, R., McCarty, W., Suárez, M. J., Todling, R., Molod, A., Takacs, L., Randles, C. A., Darmenov, A., Bosilovich, M. G., Reichle, R., Wargan, K., Coy, L., Cullather, R., Draper, C., Akella, S., Buchard, V., Conaty, A., da Silva, A. M., Gu, W., Kim, G.-K., Koster, R., Lucchesi, R., Merkova, D., Nielsen, J. E., Partyka, G., Pawson, S., Putman, W., Rienecker, M., Schubert, S. D., Sienkiewicz, M., and Zhao, B.: The Modern-Era Retrospective Analysis for Research and Applications, Version 2 (MERRA-2), J. Climate, 30, 5419–5454, https://doi.org/10.1175/JCLI-D-16-0758.1, 2017.

- Harvey, V. L., Knox, J. A., France, J. A., Fujiwara, M., Gray, L., Hirooka, T., Hitchcock, P., Hitchman, M., Kawatani, Y., Manney, G. L., McCormack, J., Orsolini, Y., Sakazaki, T., and Tomikawa, Y.: Chapter 11: Upper Stratosphere and Lower Mesosphere, SPARC Reanalysis Intercomparison Project (S-RIP) Final Report, edited by: Fujiwara, M., Manney, G. L., Gray, L. J., and Wright, J. S., SPARC Report No. 10, WCRP-6/2021, SPARC, DLR-IPA, Oberpfaffenhofen, Germany, https://doi.org/10.17874/800dee57d13, 2021. .

- Yamazaki, Y. and Matthias, V.: Large-Amplitude Quasi-10-Day Waves in the Middle Atmosphere During Final Warmings, J. Geophys. Res.-Atmos., 124, 9874–9892, https://doi.org/10.1029/2019jd030634, 2019.

---

## Author Comment (AC2)

**Response to Reviewer #1's Comments**

We are writing to provide additional details regarding the revisions made to our manuscript titled "Quasi-10-day wave activity in the southern high-latitude MLT region and its relation to the large-scale instability and gravity wave drag".

To clearly delineate the structure of our second response, the original reviewer comments are given in bold and italics, our initial responses are in grey, and the specific changes we have made are in regular red font.

**General comments:**

***Q. This study examines the generation and propagation of 10-day waves in the southern hemisphere upper atmosphere, using observations from meteor radar winds at two Antarctic stations, with supporting data from SD-WACCM simulations and MLS data. The main conclusion of this study is the identification of regions of barotropic/baroclinic instability in the upper mesosphere and lower thermosphere that can generate such oscillations***

A: The identification of regions of barotropic/baroclinic instability in the upper mesosphere and lower thermosphere as potential generators of oscillations is one of conclusions of our work. However, we would like to emphasize that our study also presents two additional key findings.

1. The first one is the climatology and seasonal variability of the quasi-10-day wave (Q10DW) observed in the Antarctica MLT region. The seasonal variation we present in our manuscript adds to the list of observational reports of mesosphere dynamics that can be used for the validation of model results as well as other observations.

2. The second one is the implication of comparison between EXP60 and EXP75. This comparison suggests that large-scale instabilities due to excessive gravity wave drag (GWD) in the summer mesosphere can make travelling quasi-10-day planetary waves (PWs) overestimated. This finding suggests that the more accurate representation of GWD is required to improve the mesosphere dynamics in high-top models.

**Major comments**

**Q1. The main conclusion of the study is not new. The authors have ignored existing literature which offers corroborating or alternative investigations: doi 10.1002/grl.50373, 1016/j.asr.2022.10.054, 10.1029/2019JD031599 and 10.1016/j.jastp.2014.06.009, and many others published since Chandran's study.**

A1: We agree that the studies you mention should be incorporated into our manuscript to provide a comprehensive background, and thus we will add these references to the introduction and discussion sections of our revised manuscript.

However, those studies particularly focus on the variation of Q10DW activity during sudden stratospheric warming (SSW) events. Our study describes the 5-year and seasonal variations of Q10DW over the Antarctic region observed using ground-based meteor radars (MRs) rather than Q10DWs during particular events. As mentioned in the introduction, Forbes & Zhang (2015) reported climatology and seasonal variations of Q10DW activity within +-50 degrees latitude in the altitude range of 20–100 km based on the SABER data for 2002–2013 . Our research also contributes to the climatology and seasonal variations at higher latitudes over the Antarctica for z= 80–100 km, Furthermore, our additional conclusion suggests that large-scale instabilities due to strong eastward GWD in the summer mesosphere can lead to the generation of Q10DW as we mentioned above.

A1': Following your suggestions, we included a detailed review of previous Q10DW studies related to SSWs. This comprehensive overview has been integrated into to the "1 introduction" section from lines 96 to 117.

**Q2. MERRA-2 is scientifically valid up to 60 km, the uppermost altitude where MLS data is used. Above 60 km and all the way to the lid (~75 km), it is just a sponge layer. Thus, I cannot associate any usefulness to the EXP75 presented in this study, where SD-WACCM is nudged with MERRA-2 fields all the way to MERRA-2 lid: it is like nudging a model with another model. Certainly, EXP60 is more useful but it is discussed only in terms of a difference from the EXP75 and briefly. If the goal of doing this study was to compare two simulations with higher and lower nudging fields, the authors should then be aware of these extant studies: doi 10.1002/2017JD027782, or 10.1002/2015GL065838.**

A2: Since late 2004, the data assimilation system at NASA GSFC GMAO have incorporated the MLS temperature and ozone from 5 hPa up to 0.02 hPa and from 250 hPa to 0.1 hPa, respectively, for the production of the MERRA-

2 data (Gelaro et al., 2017; McCormack et al..2021). We think the assimilation of MLS data gives the mesospheric meteorological analysis with a reasonable quality when compared to other mesospheric reanalysis. McCormack et al. (2021) showed the reasonable quality of the mesospheric part of the MERRA-2 data through the intercomparison of the mesospheric assimilation such as the MERRA-2 and results obtained using the higher-top data assimilation systems. The realism of the MERRA-2 in the upper mesosphere is also supported by the personal communication with Drs. Lawrence Coy and Krzysztof Wargan at NASA GSFC GMAO (8 November, 2023). They told us that there is no evidence that the divergence damping layer (Fujiwara et al., 2017) near the top boundary of the GEOS6 model is unduly interfering with the MERRA-2 results. They also said that while there is a lot of variability in the different (mesospheric) assimilation systems, meaning a lot of uncertainty, MERRA-2 seems to fit well in the mix. In fact, Figure 11.42 in Chapter 11 of the Stratosphere-troposphere Processes And their Role in Climate (SPARC) Reanalysis Intercomparison Project (S-RIP) Final Report (Harvey et al., 2021; https://www.sparc-climate.org/sparc-report-no-10) demonstrates that upper mesospheric temperature in the MERRA-2 compares well with the MLS temperature. As reviewer pointed out, there is the sponge layer near the top boundary of the GEOS6 model, but the use of the sponge does based on the divergence damping in the MERRA-2 not seem to degrade seriously the reliability of the MERRA-2 in the mesosphere. A lot of damping mechanisms (Rayleigh friction, 2nd-order hyperdiffusion, and so on) can be used for the sponge layer (Jablonowski and Williamson, 2011). Rayleigh damping and low-order hyperdiffusion can damp significantly model forecast fields, but the divergence damping is often used to effectively and selectively remove high-frequency (noisy) gravity waves keeping the large-scale circulation and planetary wave structure less changed. Below, we compare the horizontal temperature distribution between MLS and MERRA-2 data. The results demonstrate generally good agreement between the MERRA-2 and MLS above 0.1 hPa (Figure R1), suggesting that the EXP75 experiment is not simply nudged towards model forecasts but towards assimilation based on MLS. We will revise our manuscript to include a more detailed justification of this specified dynamics experiments, addressing potential concerns about the validity of simulation results in the upper mesosphere.

[Figure]

**Figure R1** Longitude-Latitude distributions of temperature at three different pressure levels of 0.03 hPa, 0.04 hPa, and 0.07 hPa. For each pressure level, the temperature distributions from (left) MLS and (right) MERRA-2 are presented.

A2': In order to clarify the top boundary of MERRA-2 reanalysis, we have included the overview of divergence damping layer for the GEOS-6 model and assimilation procedure of MERRA-2 in the 2.2 SD-WACCM section from line 201 to 210.

**Q3. MLS data is used in the data assimilation of MERRA-2. I find it odd to attempt to validate the model results that are nudged to MERRA-2 with a dataset that is part of the data assimilation cycle. In other words, an independent data set ought to be used. Why not SABER?**

A3: We agree with your opinion. Independent datasets need to be employed for validation. First of all, however,

we will check the availability of SABER data in the Antarctic region. Due to the yaw orbit of TIMES satellite, we need to confirm whether SABER data are available in the high-latitude regions for the specific dates and locations of interest. If the SABER observation is available in the Antarctic areas, we will incorporate the analysis with SABER into our manuscript to strengthen of our results.

A3': In accordance with your suggestion, we checked the availability of SABER data for the cases in Figure 6a and 6b of our study. As shown in Figure R2, we found that SABER data was not available for case '05 Feb 2013', but available for case '16 Nov 2016'. However, Figure R2b shows that the local time of observation for high-latitude regions were concentrated between 22 and 02 LT. This concentration of observations within a narrow time window, due to the yaw cycle of the TIMED satellite, makes it difficult to calculate a daily zonal mean zonal wind for specific dates. Smith et al. (2017) did calculate monthly mean zonal mean zonal wind using SABER data covering 60 days to account for the yaw cycle, this approach appears unsuitable for our study, which focuses on investigating large-scale instability on specific dates. Therefore, we concluded that using SABER data for a case study like ours might not suitable.

[Figure]

**Figure R2** Horizontal distribution of local time sampling of SABER on (a) 5 February 2013 and (b) 16 November 2016.

**Q4. The authors identify the occurrence of a 10-day normal mode from the symmetric behavior of MLS geostrophic winds. That is not sufficient: the amplitude structure has nodes, and the phase is expected to be in a specific configuration in order to be 10-day free oscillation. See: DOI: 10.1175/JAS-D-11-0103.1, 10.1111/j.1600-0870.2007.00257.x, and the Salby's papers already cited.**

A4: We agree that a rigorous analysis based on Legendre polynomials and Fast Fourier Transform, which provide latitudinal and longitudinal structures of the normal mode, is the optimal method for identifying the theoretical normal modes. However, It is noteworthy that several studies have recognized disturbance with a quasi-10-day period or other periods of gravest normal modes, westward propagation, and zonal wavenumber 1 as normal mode waves in most of the middle atmosphere (e.g., Forbes and Zhang, 2015; Yamazaki and Matthias, 2019). Forbes and Zhang (2015) observed that the normal mode of Q10DW tends to be contaminated above 80 km altitude, the layer on which our study focuses. In their study, the amplitude and latitudinal structure of Q10DW above z = 80 km are modified by the asymmetry of mean winds between the hemispheres and meridional temperature gradient. In addition, they suggested that the dissipation of gravity wave filtered by the Q10DW wind field can generate a secondary Q10DW by momentum deposition. Our supplementary figure (Fig. S3) demonstrates that this latter factor is unlikely to be a significant contributor to the generation of Q10DW in our cases. In our manuscript, we tried to distinguish Q10DW with normal mode-like structure from oscillations that seem to never have the normal mode-like structure using MLS data. As the reviewer pointed out, however, this distinction may potentially cause confusion, and it will be excluded in our revised manuscript. In the light of removing potential confusions, we will carefully define our terminology in the revised manuscript. The Q10DW will be described as "a westward-propagating (travelling) planetary wave with zonal wavenumber 1, potentially related to the Rossby normal mode". We will incorporate the considerations mentioned here to clarify our analysis and conclusions.

A4': We have modified the terminology of Q10DW as "Q10DW-W1, which is potentially related to the Rossby normal mode in the MLT region." In line 254–255.

References:

- Jablonowski, C., and Williamson, D. L.: The pros and cons of diffusion, filters and fixers in atmospheric general circulation models, in Numerical Techniques for Global Atmospheric Models, edited by: Lauritzen, P. H., Springer-Verlag, Berlin, Heidelberg, Germany, 381-493, http://doi.org/10.1007/978-3-642-11640-7_13, 2011.

- McCormack, J. P., Harvey, V. L., Randall, C. E., Pedatella, N., Koshin, D., Sato, K., Coy, L., Watanabe, S., Sassi, F., and Holt, L. A.: Intercomparison of middle atmospheric meteorological analyses for the Northern Hemisphere winter 2009–2010, Atmos. Chem. Phys., 21, 17577–17605,

https://doi.org/10.5194/acp-21-17577-2021, 2021.

- Forbes, J. M. and Zhang, X.: Quasi-10-day wave in the atmosphere, J. Geophys. Res.-Atmos., 120, 11,079–11,089, https://doi.org/10.1002/2015jd023327, 2015.

- Fujiwara, M., Wright, J. S., Manney, G. L., Gray, L. J., Anstey, J., Birner, T., Davis, S., Gerber, E. P., Harvey, V. L., Hegglin, M. I., Homeyer, C. R., Knox, J. A., Krüger, K., Lambert, A., Long, C. S., Martineau, P., Molod, A., Monge-Sanz, B. M., Santee, M. L., Tegtmeier, S., Chabrillat, S., Tan, D. G. H., Jackson, D. R., Polavarapu, S., Compo, G. P., Dragani, R., Ebisuzaki, W., Harada, Y., Kobayashi, C., McCarty, W., Onogi, K., Pawson, S., Simmons, A., Wargan, K., Whitaker, J. S., and Zou, C.-Z.: Introduction to the SPARC Reanalysis Intercomparison Project (S-RIP) and overview of the reanalysis systems, Atmos. Chem. Phys., 17, 1417–1452, https://doi.org/10.5194/acp-17-1417-2017, 2017.

- Gelaro, R., McCarty, W., Suárez, M. J., Todling, R., Molod, A., Takacs, L., Randles, C. A., Darmenov, A., Bosilovich, M. G., Reichle, R., Wargan, K., Coy, L., Cullather, R., Draper, C., Akella, S., Buchard, V., Conaty, A., da Silva, A. M., Gu, W., Kim, G.-K., Koster, R., Lucchesi, R., Merkova, D., Nielsen, J. E., Partyka, G., Pawson, S., Putman, W., Rienecker, M., Schubert, S. D., Sienkiewicz, M., and Zhao, B.: The Modern-Era Retrospective Analysis for Research and Applications, Version 2 (MERRA-2), J. Climate, 30, 5419–5454, https://doi.org/10.1175/JCLI-D-16-0758.1, 2017.

- Harvey, V. L., Knox, J. A., France, J. A., Fujiwara, M., Gray, L., Hirooka, T., Hitchcock, P., Hitchman, M., Kawatani, Y., Manney, G. L., McCormack, J., Orsolini, Y., Sakazaki, T., and Tomikawa, Y.: Chapter 11: Upper stratosphere and lower mesosphere, SPARC Reanalysis Intercomparison Project (S-RIP) final report, edited by: Fujiwara, M., Manney, G. L., Gray, L. J., and Wright, J. S., SPARC Report No. 10, WCRP-6/2021, SPARC, DLR-IPA, Oberpfaffenhofen, Germany, https://doi.org/10.17874/800dee57d13, 2021. .

- Smith, A. K., Garcia, R. R., Moss, A. C., and Mitchell, N. J.: The Semiannual oscillation of the tropical zonal wind in the middle atmosphere derived from satellite geopotential height retrievals, J. Atmos. Sci., 74, 2413–2425, https://doi.org/10.1175/jas-d-17-0067.1, 2017.

- Yamazaki, Y. and Matthias, V.: Large-Amplitude quasi-10-Day waves in the middle atmosphere during final warmings, J. Geophys. Res.-Atmos., 124, 9874–9892, https://doi.org/10.1029/2019jd030634, 2019.

---

## Author Comment (AC3)

**Response to Reviewer #2's Comments**

**Comments on "Quasi-10-day wave activity in the southern high-latitude MLT region and its relation to the large-scale instability and gravity wave drag" by Lee et al.**

**Reviewed by Yosuke Yamazaki, Leibniz Institute of Atmospheric Physics, University of Rostock**

**This study focuses on the westward-propagating quasi-10-day wave (Q10DW) in the mesosphere and lower thermosphere of the Southern Hemisphere as observed by meteor radars and simulated by the WACCM model. Two WACCM simulations were used, in one of which the model was constrained by the MERRA-2 reanalysis from the surface up to 75 km (EXP75), and in the other simulation the model was constrained similarly but up to 60 km (EXP60). After showing qualitative agreement between the Q10DWs in the meteor radar observations and EXP75 simulation, the authors examined the cause of Q10DW for selected events, and demonstrated that the barotropic/baroclinic instability played a leading role. The authors also compared the Q10DWs in EXP75 and EXP60, and noted that the wave amplitude is generally greater in EXP60. They concluded that the difference resulted from the background atmosphere that is largely controlled by gravity-wave parameterization.**

**The new findings are the sensitivity of Q10DW to the nudging range adopted in model simulations and the importance of gravity-wave drag in it. I do not have an objection for this paper to be published in Atmospheric Chemistry and Physics. However, I feel that the paper could benefit from some revisions. My specific comments can be found below.**

: Thank you for your comments concerning our manuscript "Quasi-10-day wave activity in the southern high-latitude MLT region and its relation to the large-scale instability and gravity wave drag". Those comments are valuable and helpful for improving our investigations and the importance guiding to our research. We have addressed each comment carefully and have made corrections as far as possible. The comments are in bold and our replies are in regular font. We recalculated the amplitude of Q10DW from MRs and identified certain errors in our original analysis. However, these changes do not affect the structure of our manuscript.

**Major comments:**

**Q1. Model-data comparison**

**Figures 1 and 2 show the Q10DW from meteor radars and WACCM EXP75 simulation, respectively. The disagreement is profound. The patterns of intra-annual variations are different. Also, the wave amplitudes are different by a factor of 3 or so. This makes me wonder whether the radar analysis technique has been properly applied. If yes, and if the discrepancy between the data and simulation exceeds the uncertainties in the radar results, are the simulation results still a good representation of Q10DW? How do the WACCM results compare with the MLS results? It is easier to derive Q10DW using satellite data than radar data. If there is good agreement between the WACCM and MLS results, it would make more sense to present the MLS results instead of the radar results.**

A1: We carefully examined our calculation process and identified certain errors in our original analysis. This leads to a uniform adjustment in the overall amplitude of Q10DW. However, it is important to note that the recalculated amplitude of Q10DW from MRs did not alter the seasonal variations as depicted in our manuscript's figures (Figs. 1 and 3). The recalculated amplitude is now found to be approximately one-third to one-half of those obtained from MLS and SD-WACCM. The amplitudes of Q10DW are systematically lower in MRs observations compared to SD-WACCM results. These observations agree with our conclusion that the model reproduces overestimated amplitude of Q10DW. We believe this discrepancy in the amplitude of Q10DW can be attributed to the accuracy of estimated geostrophic wind from MLS data. Figure R1 and R2 show time series of zonal and meridional winds, respectively, measured by meteor radar and geostrophic wind estimated from MLS data in the Antarctic stations (KSS: 62.2°S, 58.8°W and Davis: 68.6°S, 77.9°E) at the altitude of ~90 km. The correlation between meteor radar winds and MLS winds shows a variability of 0.3 to 0.8 depends on year. Notably, the meridional winds component shows lower correlation values than the zonal component. The lower correlation in the meridional wind is likely due to coarse longitudinal resolution of MLS, affecting the accuracy of meridional geostrophic wind estimation. Such differences in the geostrophic wind estimated from MLS data are reported by Rüfenacht et al. (2018) and may account for the observed discrepancies in amplitude of Q10DW.

While MLS data could offer insights into the global structure of Q10DW, the meteor radar observations

provide a more precise depiction of Q10DW behavior in the MLT region, especially given MLS's low vertical

resolution of about 10 km near the mesopause. Consequently, we adopted an approach begins with meteor

radar analysis to investigate local Q10DW activities in the southern high-latitude regions, followed by a

detailed analysis using SD-WACCM. Moreover, as shown in Fig. 3 of our manuscript, although the variation of

Q10DW/Q10DO amplitudes in EXP75 is not perfectly matched with those in MR for all periods, there is

significant agreement during the periods indicated by yellow boxes on the abscissa, which meet multiple

criteria as we described in the manuscript.

[Figure]

**Figure R1.** Time series of (blue and green) zonal wind measured from KSS and Davis meteor radar, respectively,

and (red and orange) zonal component of geostrophic wind estimated from MLS data near the KSS and Davis,

respectively, at the altitude of ~90 km from 2012 to 2016. Correlation coefficients (R) between the meteor radar

winds and MLS winds are shown at the top of each panel.

[Figure]

**Figure R2.** Same as Figure R1, but for meridional wind.

**Q2. EXP60 vs EXP75**

**It is stated that the amplitude of Q10DW in EXP60 is excessive (in Abstract and Summary). However, I do not see evidence to support this statement. Looking at Figures 1, 2 and S2, the amplitude of Q10DW is too small in both EXP75 and EXP60. It is difficult to say which reproduces the observations (Figure 1) better. Again, comparisons with the MLS results might help.**

A2: We apologize for any confusion. As mentioned in our response to the first major comment (Q1), we initially miscalculated the amplitude of Q10DW from MRs data, which led to an apparent higher amplitude compared to the model simulations. However, by evaluating again and recalculating these values, we found that the amplitude of Q10DW derived from MR data is actually lower than that in the EXP75, and the difference is even more pronounced when compared to EXP60.

**Q3. Introduction**

**The paragraph starting at l. 65 highlights recent studies on Q10DW. It feels that there are many studies that the authors overlooked especially in the context of the Q10DW response to sudden stratospheric warmings (e.g., Matthias et al., 2012; Yamazaki & Matthias, 2019; He et al., 2020; Wang et al., 2021; Qin et al., 2022;**

**Yin et al., 2023). It is important to address them, because some of them performed a similar analysis as the authors did in this paper and identified the importance of the barotropic/baroclinic instability.**

**Matthias et al. (2012) https://doi.org/10.1016/j.jastp.2012.04.004**

**He et al. (2020) https://doi.org/10.1029/2020GL091453**

**Qin et al. (2022) https://doi.org/10.1029/2022JD037730**

**Yin et al. (2023) https://doi.org/10.1016/j.asr.2022.10.054**

**--- only those which are currently not cited ---**

A3: Thank you for pointing out these studies. We agree that the studies you mention should be include into our manuscript to provide a comprehensive background, and thus we added these references to the introduction from line 56 to 117.

**Minor comments:**

**Q4. l. 62 "modulate the periods of tides"**
**Periods of tides would not change, as tides are defined by their periods (24h, 12h, 8h, etc.) The amplitude of tidal waves can undergo an apparent modulation due to the presence of secondary waves arising from the nonlinear interaction between planetary waves and tides.**

**Recommended reading**

**Miyoshi & Yamazaki (2020) https://doi.org/10.1029/2020JA028283**

A4: In response to your comment regarding line 62, we revised the sentence to: "In addition, the amplified PWs can interact with tidal waves through nonlinear interactions, resulting ionospheric disturbances during SSWs." This revision can be found In line 63 of the revised manuscript.

**Q5. l. 72 "high-latitude mesosphere and lower thermosphere"**

**How did they find large Q10DW in the high-latitude region while their data are limited within +/-50 latitude?**

A5: We apologize for any confusion. To clarify, we will revise the sentence line 72 from "high-latitude mesosphere and lower thermosphere" to "mid-latitude (40 – 50° latitude) mesosphere and lower thermosphere". This revision can be found In line 73 of the revised manuscript.

**Q6. l. 94 "the amplification mechanism of Q10DW-W1 still has not been investigated"**
**It may be clarified that this is about the seasonal amplification during equinoxes. As mentioned earlier, the amplification mechanism of Q10DW during sudden stratospheric warmings has been addressed in previous studies.**

A6: We are grateful for your comment. To bring more clarify to the sentence, we revised the sentence to "While the amplification mechanism of PWs generated following SSW has been addressed in previous studies (e.g., Qin et al., 2022, Yin et al., 2023), the specific mechanisms driving their seasonal amplification during equinoxes remain less explored." This revision can be found In line 118–121 of the revised manuscript.

**Q7. Figure 1**
**It would be informative if the geographical coordinates of the radars are mentioned in the figure caption.**

A7: We appreciate your suggestion. We agree that including the geographical coordinates of the meteor radars will provide valuable context to the figure. We revised the figure caption to "…meridional winds observed by MRs at Davis (68.6°S, 77.9°E) and KSS (62.2°S, 58.8°W) for 2012–2016…".

**Q8. l. 392 "F/sgn(A)" (also at l. 473)**
**What is "sgn"?**

A8: The term "sgn" refers to the sign of variable. In this context, "sgn(A)" indicates the sign of variable A (wave activity). To avoid any confusion and provide clarity, we have explicitly mentioned this in the revised manuscript. In lines 445~446, we state: The wave-activity density is shaded in blue and red depending on its sign [$\mathrm{sgn}(A)$]."

**Q9. l. 427 "6 April 2015 case"**

**Unlike the other events, the wave propagation is poleward in the stratosphere. This is a Q10DW event during a final warming event (Yamazaki & Matthias, 2019; Qin et al., 2022). Qin et al. (2022) discussed that the inter-hemispheric propagation of the wave from the Northern Hemisphere was possible because of the phase of the quasi-biennial oscillation.**

A9: We are thankful for your intriguing comment. We confirmed that our analyzed case of 6 April 2015 is included in the 2015 final warming event discussed by Yamazaki and Matthias (2019) and Qin et al. (2022). Notably, Qin et al. (2022) reported that the averaged meridional component of Eliassen-Palm Flux from 18 March to 18 April 2015 extends from the stratosphere in the Northern-Hemisphere across the equator to the stratosphere in the Southern Hemisphere (Qin et al., 2022). This finding is consistent with our results. In addition, Figure 9b and 9c in Qin et al. (2022) suggests that the phase of the semi-annual oscillation is also in its westerly phase. This observation, in conjunction with the westerly phase of the QBO in the middle stratosphere, supports the southward propagation of Q10DW from Northern Hemisphere in a wide altitude range from middle to upper stratosphere. To enhance our manuscript, we included additional explanations and the relevant references to this particular aspect in lines 497~502.

**Q10. l. 432 "divergent"**

**It should be "divergence".**

A10: We will change the text accordingly. This revision can be found In line 494 of the revised manuscript.

**Q11. l. 463 "s>=20"**

**It is strange that only GWs with high zonal wavenumbers are resolved. Maybe "s<=20"?**

A11: As shown in Figs. S3 (b) and (c), our analysis considered two potential contributors to the amplification of Q10DW: 1) nonconservative gravity wave drag (NCGWD) due to parameterized GWD with a quasi-10-day periodicity, and 2) resolved GWs with quasi-10-day periodicity, specifically those with higher zonal wavenumbers ($s>=20$, shorter wavelengths). Our investigation revealed that both NCGWD due to parameterized GWD with a quasi-10-day periodicity and EPD generated by resolved GWs with quasi-10-day periodicity are relatively weak. Therefore, we infer that the Q10DW are likely generated in-situ due to the large-scale instability rather than being a result of GWD. To further clarify, we have added the following sentence to lines 533~535 in revised manuscript:" … In addition, Forbes and Zhang (2015) suggested that the dissipation of gravity waves filtered by the Q10DW wind field can generate a secondary Q10DW by momentum deposition. In this regard, the both parameterized GWs and resolved GWs…"

**Q12. l. 483 "February and November"**

**Which year?**

A12: As shown in Fig. S2, we found that in all yeas except 2014, the amplitude of Q10DW in EXP60 is stronger than in EXP75 during the February and November. Thus, the reference to "February and November" in our manuscript is not meant to indicate a specific year but rather a consistent pattern revealed in multiple years. To avoid any confusion, we will modify the sentence to "This section compares the Q10Dos around the mesospheric instability regions in the two SD-WACCM simulations (EXP75 and EXP60) during February and November across multiple years."

**Q13. l. 484 "February and November are chosen because ..."**

**The justification is weak. It would make more sense to select the events that are examined in Figure 4,**

**where the Q10DWs are not only substantial in amplitude but also in qualitative agreement with observations.**

A13: We conducted the analysis of all February and November cases as indicated by the yellow shaded date in Fig. S2. This analysis revealed that the wave activity of Q10DW in EXP60 is consistently more pronounced compared to EXP75 across all cases as shown in Figure R3. In the manuscript, we specifically chose to highlight cases of 5 February 2013 and 16 November 2016, as examples. To support our analysis, we will also include Figure S4 in the supplementary, which is similar to Figure R3.

[Figure]

**Figure R3.** EP flux parallel to local group velocity [$\mathbf{F}/\mathrm{sgn}(A)$] and normalized wave activity density [$A$ $(\rho_0 a \cos \phi)^{-1}$ given in the unit of m s$^{-1}$] for the Q10DWs on (a and b) 3 February 2012, (c and d) 29 September 2012, (e and f) 18 October 2012, (g and h) 17 November 2012, (I and j) 2 November 2013, and (k and l) 25 November 2015. The first and third columns and second and forth columns represent the results from EXP75 and EXP60, respectively. The activity density $A$ is shaded in blue and red depending on its sign. The boundaries of the instability regions ($\overline{q}_\phi = 0$, green lines), the negative $n^2$ regions (grey shading), and the red contours for zonal-mean zonal wind are overplotted. For

eastward (westward) zonal-mean zonal wind, contours are plotted in solid (dashed) lines, and contour interval is 10 m s$^{-1}$.

**Q14. Summary**

**I do not see the merit of listing items here (#1-#6). #2 and #5 are not the description of results but the description of what the authors did to get the results.**

A14: Thank you for your comment regarding the structure of the summary section. In response, we revised the summary to adopt a narrative style. This approach will ensure a more integrated presentation of both our methodologies and results.

References:

Forbes, J. M. and Zhang, X.: Quasi-10-day wave in the atmosphere, J. Geophys. Res.-Atmos., 120, 11,079–11,089, https://doi.org/10.1002/2015jd023327, 2015.

Rüfenacht, R., Baumgarten, G., Hildebrand, J., Schranz, F., Matthias, V., Stober, G., Lübken, F.-J., and Kämpfer, N.: Intercomparison of middle-atmospheric wind in observations and models, Atmos. Meas. Tech., 11, 1971–1987, https://doi.org/10.5194/amt-11-1971-2018, 2018.

Qin, Y., Gu, S. Y., Dou, X., Teng, C. K. M., Yang, Z., and Sun, R.: Southern Hemisphere Response to the Secondary Planetary Waves Generated During the Arctic Sudden Stratospheric Final Warmings: Influence of the Quasi-Biennial Oscillation, J. Geophys. Res.-Atmos., 127(24), e2022JD037730, https://doi.org/10.1029/2022JD037730

Yamazaki, Y. and Matthias, V.: Large-amplitude quasi-10-day waves in the middle atmosphere during final warmings, J. Geophys. Res.-Atmos., 124, 9874–9892, https://doi.org/10.1029/2019jd030634, 2019.

Yin, S., Ma, Z., Gong, Y., Zhang, S., and Li, G.,: Response of quasi-10-day waves in the MLT region to the sudden

stratospheric warming in March 2020, Adv. Space. Res., 71(1), 298–305,

https://doi.org/10.1016/j.asr.2022.10.054

---

## Author Response (AR2)

**Response to RC1: 'Comment on egusphere-2023-2381', Anonymous Referee #1, 21 Jan 2024**

We are very pleased with RC1's comments. The revier comments are given in italics and the our repective responses in purple font.

*"I appreciate the effort of the authors to address the comments from this reviewer; I realize it wasn't easy. I remain concerned of the use of the model output to infer climatological behavior in a region that is not informed by observations and is potentially affected by lid-dynamics. The attempt to 'validate' the model in their rebuttal (figure R1) addresses only a static behavior of the model, not its variability. I only ask that the authors include in their conclusions a cautionary statement that the behavior illustrated in their study might be influenced by compounded effects of the sponge layer dynamics and unconstrained model climatology. After that, the study can proceed to publication."*

: In response to your concerns regarding the model output potentially being affected by the sponge layer, we have included a cautionary note in our manuscipt. This can be found in lines 694–703 as follows:

Lines 694–703: "Results of SD-WACCM may depend on the extra damping above the middle mesosphere in the GEOS-6 model (Fujiwara et al., 2017) used to produce the MERRA-2 data. The damping may have harmful effects on the results for the upper mesosphere in the EXP75, where the dynamics is still specified above the middle mesosphere using the MERRA-2, but comparison with observations shows that the zonal asymmetric structure of mesospheric temperature in the EXP75 is reasonable for the time periods of our interest (Fig. S5). However, the activity and variability of mesospheric PWs in the MERRA-2 and SD-WACCM need to be further examined for the longer time periods and evaluated against other observations to support the reliability of results obtained in this study, which should be a topic of continuing research."

We believe that this addition adequately addresses your concerns about the potential influence of sponge layer dynamics on our results.

**Response to RC2: 'Comment on egusphere-2023-2381', Yosuke Yamazaki, 26 Jan 2024**

We appreciate the comments by Dr. Yamazaki. We repeat the reviewer's comments are provided in italics and the our repective responses in purple font.

*"I have read through the revised manuscript and response letter. The authors have properly addressed my previous comments.*

*The authors have re-evaluated the amplitude of the Q10DW derived from the meteor radar (MR) observations. The Q10DW amplitudes have been corrected from 10-25 m/s to 1.2-3.0 m/s. The amplitude in the WACCM simulation (EXP75) is in the range of 4-10 m/s, which is much larger than the revised amplitude values. The Q10DW amplitudes estimated with MLS geostrophic winds are also greater than those from the MR observations. The authors argue that the Q10DWs from WACCM and MLS are overestimated. The authors explained in the response letter why this might be the case. Although I have no objection or counter evidence to the authors' viewpoint, it is worth acknowledging and discussing the disparities among the results obtained from MR, MLS, and WACCM in the paper. For the moment, it is not entirely clear which among these three sources most accurately represents the true nature of the Q10DW. The MR technique is not perfect, involving only two stations at different latitudes. More validation studies are needed in the future, which may lead to the improvement of the technique as well as the establishment of a method for uncertainty estimate.*

*I suggest that the authors include one sentence in Results and Discussion addressing the possible underestimation of the Q10DW amplitude by the MR technique, and another sentence in Summary that explicitly address the discrepancies among the MR, MLS, and WACCM results and the need for further studies."*

: In response to your suggestions, we have added contents to our manuscript discussing the potential reaseons for the discrepancies in the amplitudes of Q10DW among MRs, MLS, and SD-WACCM data as follows:

Lines 276–280: "It is important to note that the amplitudes of Q10DW are systematically lower in MRs compared to the MLS results. These discrepancies might be attributed to the accuracy of estimated geostrophic winds from the MLS data, or the inherent limitations of MR analysis, which in our case involves only two stations located at slightly different latitudes."

Lines 673–678: "In addition, our study shows the Q10DWs from the MLS appear to be consistently overestimated compared to those from MRs. These discrepancies can be due to both errors in estimating winds from the MLS and uncertainties in results obtained from two MR stations alone. Further investigation is required for more reliable estimation of the amplitude and phase of Q10DWs from observations."

These additions to our manuscript provide the possible reasons of discrepancies among the MR, MLS, and model results and emphasize the need for further studies.

Minor comments:

1. l. 97 "(2022a, 2022b)"
"(2020a, 2020b)"

: We have changed it in line 97.

2. l. 303 "generally larger"
"much smaller"

: We have updated this in line 289.